# Ca$^{2+}$-activated Cl$^-$ channel TMEM16A/ANO1 identified in zebrafish skeletal muscle is crucial for action potential acceleration

Anamika Dayal [1], Shu Fun J. Ng[1] & Manfred Grabner [1]

The Ca$^{2+}$-activated Cl$^-$ channel (CaCC) TMEM16A/Anoctamin 1 (ANO1) is expressed in gastrointestinal epithelia and smooth muscle cells where it mediates secretion and intestinal motility. However, ANO1 Cl$^-$ conductance has never been reported to play a role in skeletal muscle. Here we show that ANO1 is robustly expressed in the highly evolved skeletal musculature of the euteleost species zebrafish. We characterised ANO1 as *bonafide* CaCC which is activated close to maximum by Ca$^{2+}$ ions released from the SR during excitation-contraction (EC) coupling. Consequently, our study addressed the question about the physiological advantage of implementation of ANO1 into the euteleost skeletal-muscle EC coupling machinery. Our results reveal that Cl$^-$ influx through ANO1 plays an essential role in restricting the width of skeletal-muscle action potentials (APs) by accelerating the repolarisation phase. Resulting slimmer APs enable higher AP-frequencies and apparently tighter controlled, faster and stronger muscle contractions, crucial for high speed movements.

[1] Department of Medical Genetics, Molecular and Clinical Pharmacology, Division of Biochemical Pharmacology, Medical University of Innsbruck, Peter Mayr Strasse 1, A-6020 Innsbruck, Austria. These authors contributed equally: Anamika Dayal, Shu Fun J. Ng. Correspondence and requests for materials should be addressed to A.D. (email: anamika.dayal@i-med.ac.at) or to M.G. (email: manfred.grabner@i-med.ac.at)

Excitation-contraction (EC) coupling in vertebrate skeletal muscle is initiated at the neuromuscular junction by an action potential (AP) from a single somatic efferent motor neuron. This neuronal AP causes the release of acetylcholine which binds to nicotinic acetylcholine receptors in the motor endplate of the muscle fibre, inducing influx of $Na^+$ ions. This in turn, causes a depolarising excitatory postsynaptic potential which as soon exceeds a certain threshold level, triggers a sarcolemmal AP. Subsequently, these depolarisations travel downwards into specific sarcolemmal invaginations, the transvers (t)-tubules, where they are detected by the voltage-sensing $\alpha_{1S}$ subunit of the dihydropyridine receptor (DHPR). This induces a conformational change in the DHPR which, via allosteric coupling, is transduced to the sarcoplasmic $Ca^{2+}$ release channel or ryanodine receptor type-1 (RyR1)[1]. Opening of RyR1 leads to a massive release of $Ca^{2+}$ ions from sarcoplasmic reticulum (SR) stores into the cytoplasmic gap[2,3] of this triadic junction, which bind to troponin C of the thin filaments, finally inducing muscle contraction via actin-myosin cross-bridge interactions.

Previous studies on EC coupling in zebrafish (Danio rerio) skeletal muscle revealed that the skeletal musculature of euteleost fishes is evolutionary highly advanced compared to mammals[4,5]. Similar to mammals, teleost skeletal muscles are composed of two major types of muscle fibres, classified as slow- (type I) and fast-twitch (type II) fibres. However, in contrast to mammals, where slow- and fast-twitch fibres are intermingled, teleost axial musculature displays a clear separation, with slow (oxidative/red) muscles found on the lateral surface and fast (glycolytic/white) muscles forming the deeper layers. Additionally, the selective expression of each of the two isoforms of DHPRα$_{1S}$ subunit[4] and RyR1[6] in red and white zebrafish skeletal muscles resulted in the formation of two different muscle-type specific DHPR-RyR1 couplons. Again in contrast, mammals express only one type of couplon, regardless if it is in type I or type II fibres. This couplon-specification in zebrafish is just one example of how the 3rd round of teleost-specific whole-genome duplication (Ts3R) at the basis of the teleost clade[7,8] resulted in isoform formation during phylogenetic organ differentiation and hence in physiological specialisation[4,5]. Ts3R was also the initiation point of the loss of $Ca^{2+}$ conductance through both the DHPRα$_{1S}$ isoforms[5], interestingly, by involving distinct point mutations[4]. Notably, our investigations on the zebrafish skeletal muscle EC coupling machinery enabled us to detect another intriguing physiological and biophysical difference in EC coupling between euteleost and mammalian skeletal muscle—the unexpected participation of a $Ca^{2+}$-activated $Cl^-$ channel (CaCC).

CaCC currents are involved in multiple physiological processes, ranging from sensory transduction[9], epithelial secretion[10], to smooth muscle contraction[11]. CaCC opening in smooth muscle cells results in membrane depolarisation due to $Cl^-$ efflux, since in contrast to skeletal muscle cells the intracellular $Cl^-$ concentration in smooth muscle cells is high due to active accumulation by $Cl^-/HCO_3^-$ exchange and $Na^+$, $K^+$, $Cl^-$ co-transportation[11]. Smooth muscle CaCCs are activated by different sources of intracellular $Ca^{2+}$ increase, viz. via $Ca^{2+}$ entry through voltage-gated $Ca^{2+}$ channels, $Ca^{2+}$ release from intracellular stores as a result of G-protein-coupled pathways, or $Ca^{2+}$-induced $Ca^{2+}$ release through ryanodine receptors. Despite their crucial role in a wide range of biological processes, the molecular identity of CaCCs remained elusive until the year 2008, when three independent research groups[12–14] identified 'Transmembrane protein with unknown function 16' (TMEM16A), which is part of the 10-member mammalian TMEM16 family[15], to be accountable for CaCC currents. Hydropathy analysis indicated that the anion-selective channel embraces 8 (octa) transmembrane helices with cytosolic C- and N-terminal ends, and hence

was referred to as anoctamin 1 (ANO1)[14]. In general, ANO proteins are homo-dimers[16,17], and in the case of ANO1 each subunit is activated independently[18,19], suggesting a double-barrelled channel similar to the ClC $Cl^-$ channel[20]. Newer reports, based on crystal structure analyses and cryo-electron microscopy, proposed that the membrane-spanning domain of ANO1 consists of 10 transmembrane α-helices[21,22]. Hallmark biophysical features of ANO1 currents are voltage- and $Ca^{2+}$-dependent activation[23,24] with an outward rectification at lower micromolar $Ca^{2+}$ concentrations and a linear current–voltage (I–V) relationship at higher $Ca^{2+}$ concentrations[25,26].

In the present study, we identify and characterise ANO1 expression pattern, subcellular distribution, and current properties in zebrafish skeletal muscle by implementing molecular, immunocytochemical, and biophysical approaches. Experiments on myotubes derived from the DHPRβ$_1$-null zebrafish mutant relaxed which lack SR $Ca^{2+}$ release[27], reveal that $Cl^-$ influx via sarcolemmal ANO1 channels is activated by $Ca^{2+}$ ions released through ryanodine receptors. Interestingly, our results demonstrate that this ANO1-mediated $Cl^-$ influx accelerates the repolarisation phase and thereby decreases the duration of the skeletal muscle AP, apparently acting synergistically with the canonical $K^+$ efflux repolarisation mechanism. Furthermore, we show that evolution of this accelerated AP repolarisation mechanism in the euteleost species Danio rerio allows enhanced muscle stimulation frequencies during AP trains, consequently expected to generate tighter muscle control as well as increased force production[28–32]—an all over improvement in muscle properties especially vital for the aquatic pray-predator environment.

## Results

### SR $Ca^{2+}$ release awakes zebrafish skeletal muscle $Cl^-$ current.

Patch-clamp recordings from freshly dissociated zebrafish skeletal myotubes (Fig. 1a) displayed a very distinct picture compared to mouse myotubes, even under identical experimental conditions[30,33]. Although the standard depolarisation protocols elicited the expected robust SR $Ca^{2+}$ release in zebrafish myotubes, the archetypal slow DHPR $Ca^{2+}$ inward current was missing[4], and surprisingly was 'replaced' by a huge outward current (Fig. 1a). To test if this outward current could be massive $Cl^-$ influx, we measured whole-cell currents under $Cl^-$ free conditions[34]. And indeed, under $Cl^-$-free conditions the outward current nearly extinguished whereas SR $Ca^{2+}$ release remained intact ($P > 0.05$) (Fig. 1b). To assess if this $Cl^-$ current was $Ca^{2+}$-dependent or more precisely dependent on $Ca^{2+}$ released from the SR during EC coupling, we repeated the recordings under standard $Cl^-$ conditions (165 mM $Cl^-$ in external and 4 mM $Cl^-$ in internal solution), but on myotubes isolated from the DHPRβ$_1$-null zebrafish mutant relaxed, which lacks skeletal muscle EC coupling[27]. As demonstrated in Fig. 1c, d the lack of considerable $Cl^-$ current (left panels) is concordant with the lack of SR $Ca^{2+}$ release (right panels). Overall, integrating the above results we identified this molecular participant of the zebrafish skeletal muscle EC coupling mechanism as a $Ca^{2+}$-activated $Cl^-$ channel (CaCC). The voltage dependence of the zebrafish skeletal muscle CaCC current (Fig. 1d) strongly resembles Anoctamin (ANO) currents described in mammalian smooth muscles[35,36].

### Zebrafish mutant relaxed contains functional CaCCs.

As a first step, to validate if zebrafish relaxed myotubes are a suitable system for studying skeletal muscle CaCCs, we tested the physiological availability of CaCCs in relaxed myotubes by SR store depletion experiments in the presence of the RyR1 agonist

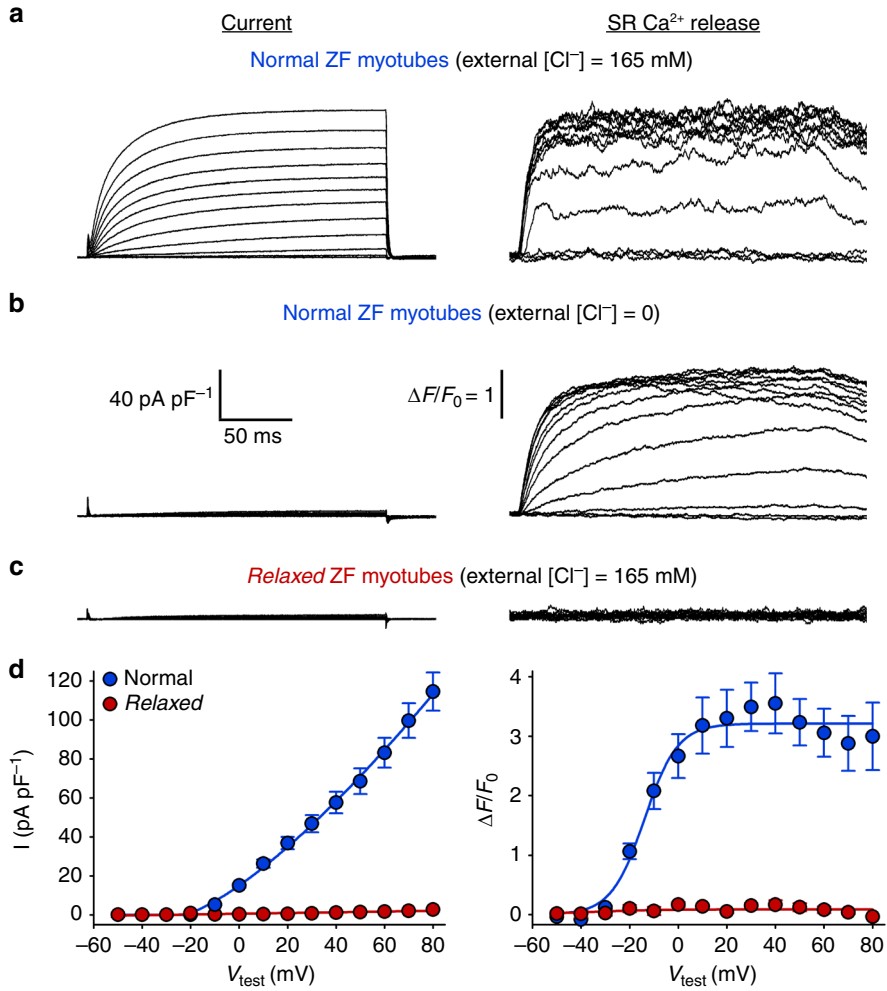

**Fig. 1** SR $Ca^{2+}$ release activates $Cl^-$ currents in zebrafish skeletal muscle. **a–c** Representative recordings from zebrafish myotubes elicited by 200-ms depolarising test potentials between $-50$ and $+80$ mV. **a** Robust outward currents (left traces) and intracellular SR $Ca^{2+}$ release (right traces) were recorded from normal myotubes under standard external solution containing 165 mM $Cl^-$. **b** In contrast, under $Cl^-$ free conditions normal myotubes displayed only marginal currents ($I = 2.99 \pm 0.20$ pA $pF^{-1}$ at $+80$ mV, $n = 5$) (left traces) but unaltered ($P > 0.05$) SR $Ca^{2+}$ release (($\Delta F/F_0)_{max} = 3.01 \pm 0.24$, $n = 5$) (right traces). Scale bars, (left) 50 ms (horizontal), 40 pA $pF^{-1}$ (vertical); (right) $\Delta F/F_0 = 1$ (vertical). **c** Representative recordings from *relaxed* zebrafish myotubes with standard external solution containing 165 mM $Cl^-$ showed neither considerable outward currents (left traces) nor SR $Ca^{2+}$ release (right traces), identifying this outward current as SR $Ca^{2+}$-release-activated $Cl^-$ current. **d** Left graph, plots of current–voltage relationship under standard $Cl^-$ conditions (165 mM) from normal ($I = 114.55 \pm 9.81$ pA $pF^{-1}$ at $+80$ mV, $n = 5$) and *relaxed* myotubes ($I = 2.70 \pm 0.02$ pA $pF^{-1}$ at $+80$ mV, $n = 5$), and (right graph) voltage dependence of maximal $Ca^{2+}$ release from normal (($\Delta F/F_0)_{max} = 3.23 \pm 0.45$, $n = 5$) and *relaxed* (($\Delta F/F_0)_{max} = 0.11 \pm 0.05$, $n = 5$) myotubes. Data are presented as mean $\pm$ s.e.m.; $P$ determined by unpaired Student's $t$-test

caffeine, using the pulse protocol depicted in Fig. 2a. As expected, even before application of caffeine we recorded robust CaCC outward currents at $+40$ mV and smaller inward currents at $-120$ mV from normal control myotubes, but very marginal currents from *relaxed* myotubes (Fig. 2b). We observed a pronounced current rundown in normal myotubes (Fig. 2d), which is a characteristic of ANO currents and is most probably produced by phosphorylation of the channel by $Ca^{2+}$/calmodulin-dependent protein kinase II (CamKII)[37,38].

Notably, application of 8 mM caffeine after the 14th sweep of the pulse protocol (Fig. 2a) induced comparable ($P > 0.05$) augmentation of CaCC currents, in both normal as well as *relaxed* myotubes (Fig. 2c, d). These results unambiguously demonstrate that *relaxed* myotubes (i) contain intact SR $Ca^{2+}$ stores and (ii) express functional CaCCs which can be activated by intracellular $Ca^{2+}$ ions. Thus, zebrafish *relaxed* myotubes serve as an ideal experimental system for studying CaCCs under SR $Ca^{2+}$-release free conditions.

**Two ANO1 isoforms are expressed in zebrafish skeletal muscle.** Since strict $Ca^{2+}$-dependence (Fig. 1c, d; Fig. 2), nearly-linear outward voltage dependence (Fig. 1d, left graph), and pronounced current rundown (Fig. 2d) are characteristics for $Cl^-$ currents of the ANO channel family, the apparent question arose which ANO isoform(s) is/are expressed in zebrafish skeletal muscle. Out of the 10 isoforms of the ANO protein family, only ANO1 and ANO2 are verified CaCCs (Supplementary Fig. 1a), while the other 8 ANO proteins, either work as scramblases or have unknown functions[39,40]. As a result of the Ts3R genome duplication[7,8] zebrafish has two ANO1 and two ANO2 genes. The pedigree in Fig. 3a, with the *Drosophila* ANO protein as outgroup, indicates that the predicted translational products of the two ANO1 and two ANO2 zebrafish isoforms cluster well with their respective mouse counterparts but show a higher degree of phylogenetic advance.

ANO1 and ANO2 isoform-specific RT-PCR primers were designed (Supplementary Table 1) to amplify DNA fragments

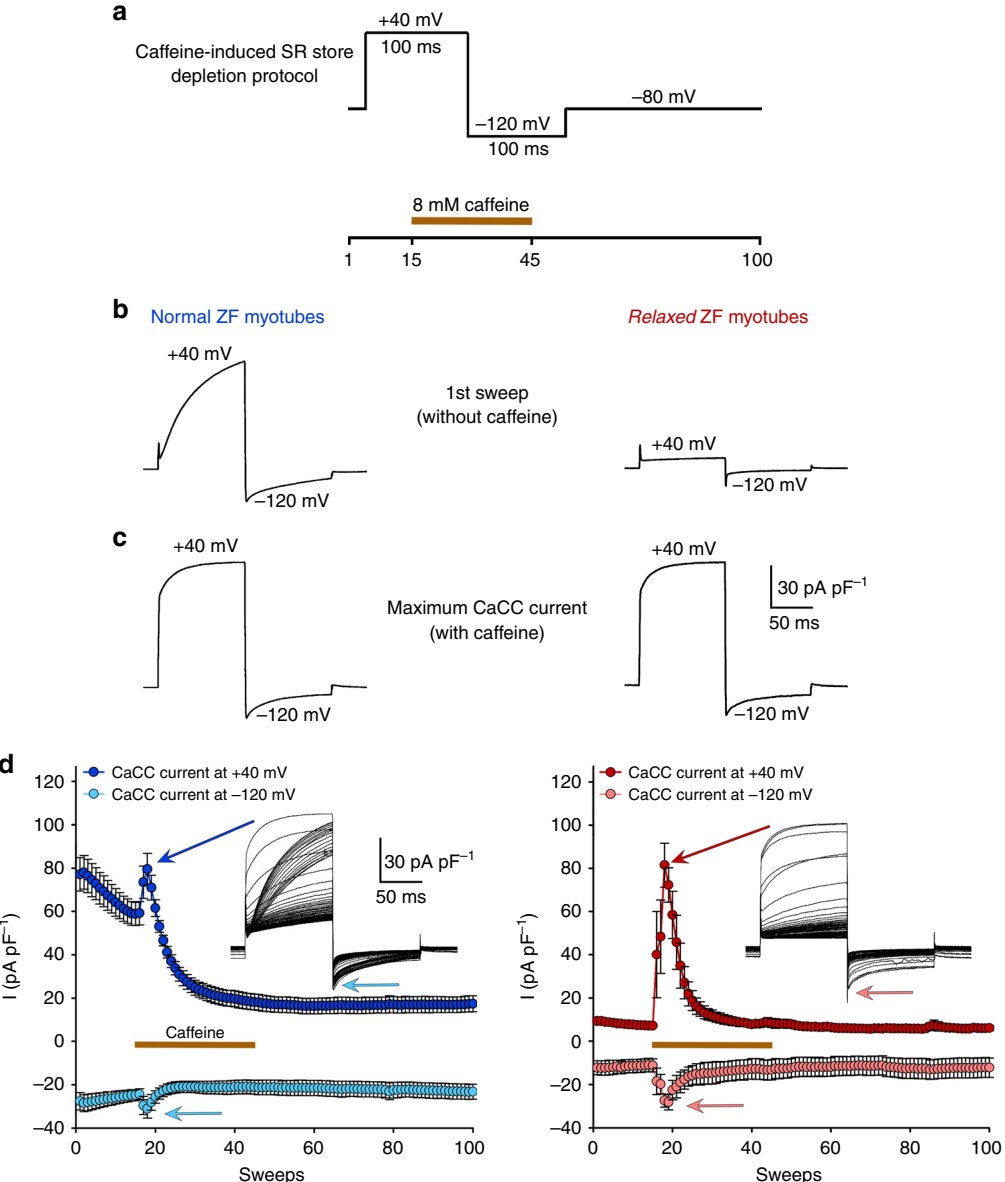

**Fig. 2** *Relaxed* myotubes express functional CaCCs and contain intact SR Ca$^{2+}$ stores. **a** Test-pulse protocol from a holding potential of −80 mV to +40 mV for 100 ms followed by a 100-ms pulse to −120 mV. The entire protocol was repeated 100 times. Brown bar indicates the application of 8 mM caffeine to induce SR Ca$^{2+}$ store depletion between 15th and 45th sweep, which was terminated by perfusion with caffeine-free standard bath solution. **b** Representative sweep of CaCC currents before application of caffeine showing robust CaCC outward current at +40 mV ($I_{max} = 77.12 \pm 7.69$ pA pF$^{-1}$, $n = 5$) and smaller inward current at −120 mV ($I_{max} = -27.85 \pm 3.99$ pA pF$^{-1}$, $n = 5$) from normal control myotubes (left trace), in contrast to only marginal outward ($P < 0.001$) and inward ($P < 0.05$) currents ($I_{max} = 9.32 \pm 0.97$ and $-12.27 \pm 3.20$ pA pF$^{-1}$, $n = 5$, respectively) from *relaxed* myotubes (right trace). **c** Representative sweep of CaCC currents after application of 8 mM caffeine, showing indistinguishable ($P > 0.05$) CaCC currents at +40 and −120 mV between normal ($I_{max} = 79.50 \pm 7.29$ and $-31.12 \pm 4.22$ pA pF$^{-1}$, respectively, $n = 5$) (left trace) and *relaxed* ($I_{max} = 81.42 \pm 10.14$ and $-28.18 \pm 3.52$ pA pF$^{-1}$, respectively, $n = 5$) (right trace) myotubes. Scale bars, 50 ms (horizontal), 30 pA pF$^{-1}$ (vertical). **d** Plots of CaCC currents vs. sweeps at +40 and −120 mV with corresponding representative CaCC current traces (insets) from normal (left graph) and *relaxed* (right graph) myotubes. After caffeine application, the maximal CaCC outward current at +40 mV is indicated by dark blue or dark red arrows and the maximal inward current at −120 mV by light blue or light red arrows, from normal or *relaxed* myotubes, respectively. Data are presented as mean ± s.e.m.; $P$ determined by unpaired Student's $t$-test

coding for a region between transmembrane α-helices 4 and 9 containing amino acid residues N650, E654, E702, E705, E734, and D738, supposed to delineate the ANO Ca$^{2+}$-binding site[21,41]. Supplementary Figure 1b depicts the positions of these six residues on the proposed 10-helix membrane folding model of ANO channels[21]. Amino acid sequence alignment of ANO1 isoforms of zebrafish and mouse shows conservation of these six residues in both the ANO1 isoforms of zebrafish (Supplementary

Fig. 1c). PCR amplification of ANO1 and ANO2 isoforms from first strands of whole adult zebrafish confirmed the accuracy of the four primer pairs and amplified fragments from all targeted ANO isoforms (Fig. 3b) to a similar extent ($P > 0.05$). On the contrary, RT-PCR amplification of ANO1 and ANO2 isoform fragments with first strands from red and white skeletal musculature showed no signals for both ANO2 isoforms (Fig. 3c) but signals for both ANO1 isoforms (Fig. 3d). More specifically,

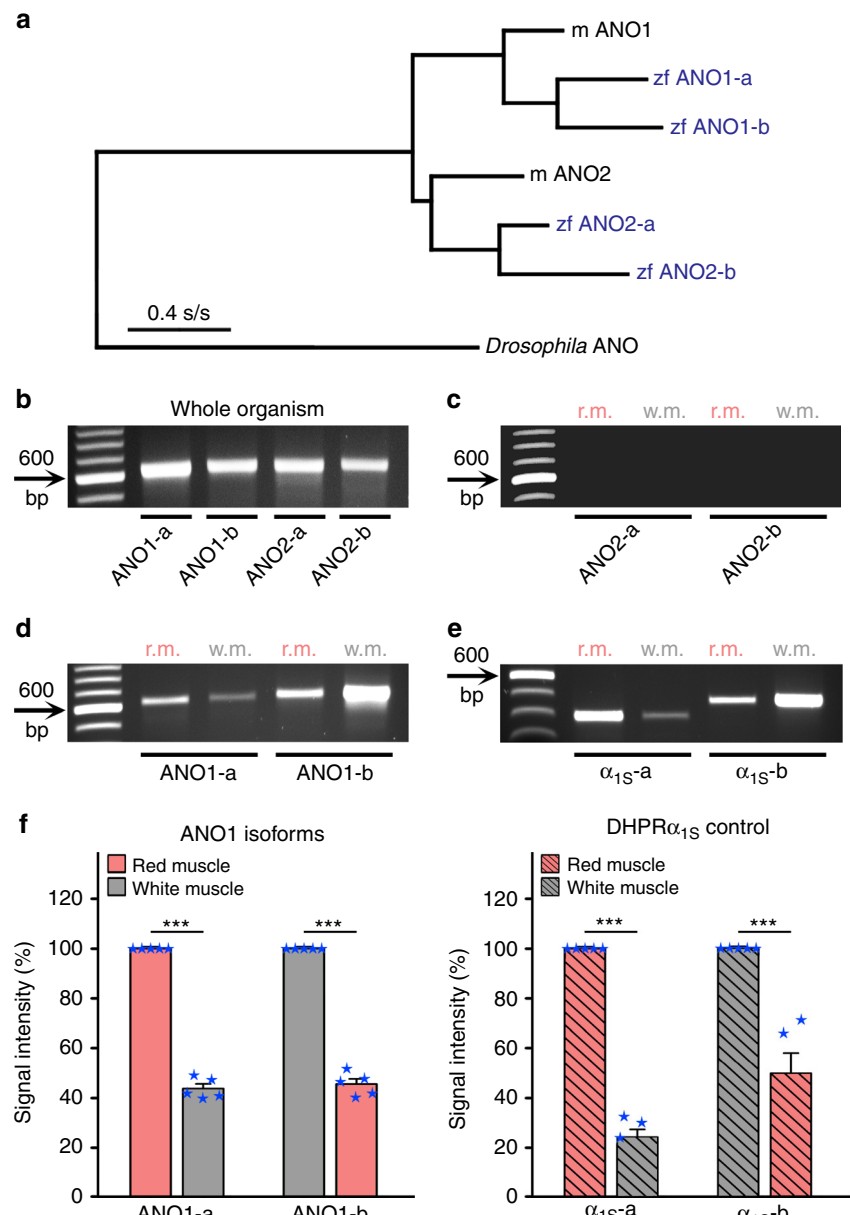

**Fig. 3** Two distinct ANO1 isoforms are expressed in red and white skeletal muscle of zebrafish. **a** Likelihood-based phylogenetic tree (Treefinder, version 2011) from aligned amino acid sequences of mouse (m) ANO1 (GenBank accession no. NP_848757), zebrafish (zf) isoforms ANO1-a (accession no. XP_009301746) and ANO1-b (accession no. XP_017209649), mouse ANO2 (accession no. NP_705817), zebrafish isoforms ANO2-a (accession no. XP_009298253) and ANO2-b (accession no. XM_009293432), with the ancestral *Drosophila* ANO channel (accession no. NP_648535) as outgroup, shows species-independent isoform clustering. Likelihood score, −16,906.7; edge lengths optimised. Scale bar, evolutionary distance in substitutions per site (s/s). **b** RT-PCR amplification from first strands of whole adult zebrafish with primers specific for all 4 ANO channel isoforms (Supplementary Table 1) showed signals with similar intensity ($P > 0.05$, $n = 5$) for RNAs that were isolated from adult normal zebrafish. **c** RT-PCR amplification with ANO2-specific primers under identical PCR conditions did not show any signal, neither from red muscle (r.m.) nor white muscle (w.m.). **d** In contrast, ANO1-specific primers amplified PCR fragments from both muscle types but to different extents. **e** Similarly, control PCR amplifications with DHP$\alpha_{1S}$-specific primers detected both, $\alpha_{1S}$-a and $\alpha_{1S}$-b in both tissues but again to different extents. **f** Left graph, quantification of band intensities of ANO1 isoforms revealed that ANO1-a is predominantly expressed in the red muscle compared ($P < 0.001$) to white muscle ($43.75 \pm 1.89\%$, $n = 5$), and ANO1-b is mainly expressed in the white muscle compared ($P < 0.001$) to red muscle ($45.56 \pm 2.08\%$, $n = 5$). Right graph, DHPR$\alpha_{1S}$-a and $\alpha_{1S}$-b, known to be exclusively expressed in red and white muscles, respectively[4], displayed a similar cross-contamination pattern for DHPR$\alpha_{1S}$-a in white ($24.21 \pm 3.12\%$; $n = 5$) and DHPR$\alpha_{1S}$-b in red muscle ($49.90 \pm 8.09\%$; $n = 5$). Bars represent mean ± s.e.m. and overlaying blue stars indicate individual data points; *** $P < 0.001$ determined by unpaired Student's *t*-test

ANO1-a isoform-specific primers predominantly amplified the DNA fragment from red muscle and ANO1-b-specific primers from white muscle. Furthermore, RT-PCR amplification by using two additional ANO2 pan-primer pairs confirmed the non-existence of ANO2 isoforms in skeletal muscle (Supplementary

Fig. 2). In order to test the purity of the tissue sample preparations, positive control PCRs with DHPR$\alpha_{1S}$-a and DHPR$\alpha_{1S}$-b isoform-specific primers and identical first strands were performed (Fig. 3e). Since $\alpha_{1S}$-a and $\alpha_{1S}$-b isoforms are exclusively expressed in red and white muscle, respectively[4], the

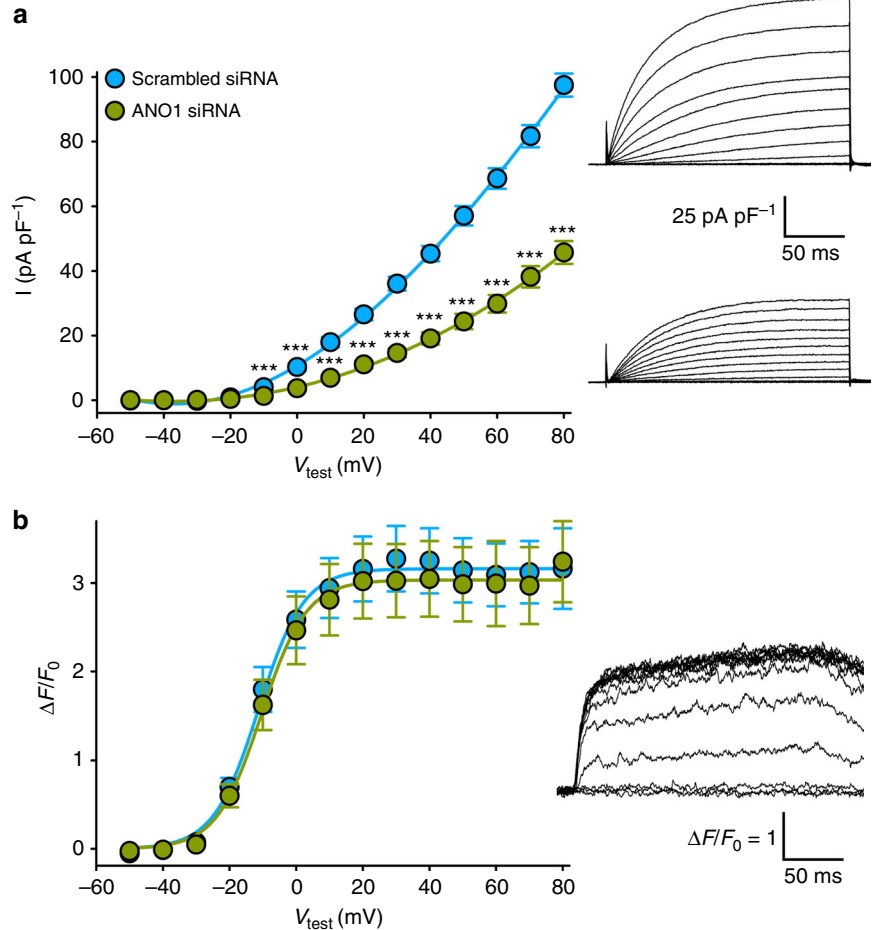

**Fig. 4** ANO1 isoforms are accountable for CaCC conductance in zebrafish skeletal muscle. **a** Plots of current–voltage relationship from normal myotubes expressing control scrambled siRNA ($I = 97.47 \pm 3.59$ pA pF$^{-1}$ at $+80$ mV, $n = 28$) or ANO1-specific (nt position 2073–2093; ANO1-b numbering) siRNA ($I = 45.71 \pm 3.52$ pA pF$^{-1}$ at $+80$ mV, $n = 34$) showing highly significant ($P < 0.001$) siRNA knock-down effects between test potentials of $-10$ to $+80$ mV. Representative recordings of corresponding ANO1 currents (250-ms pulse protocol) are depicted on the right. Scale bars, 50 ms (horizontal), 25 pA pF$^{-1}$ (vertical). **b** Plots of voltage dependence recordings of maximal Ca$^{2+}$ release from normal myotubes expressing control scrambled siRNA (($\Delta F/F_0$)$_{max}$ = $3.12 \pm 0.35$, $n = 16$) or ANO1-specific siRNA (($\Delta F/F_0$)$_{max}$ = $3.18 \pm 0.40$, $n = 16$). Representative SR Ca$^{2+}$ release recording from a myotube expressing ANO1-specific siRNA is depicted on the right. Scale bars, 50 ms (horizontal), $\Delta F/F_0 = 1$ (vertical). Bars represent mean $\pm$ s.e.m.; ***$P < 0.001$ determined by unpaired Student's $t$-test

respective weaker amplification products in Fig. 3d, e are merely due to cross-contaminated red and white muscle preparations from the relatively small-sized zebrafish. Hence, from the very similar amplification intensity profile of ANO1 and DHPRα$_{1S}$ isoforms (Fig. 3f) we can conclude that ANO1-a is predominantly expressed in the red and ANO1-b in the white skeletal musculature of zebrafish.

To confirm that ANO1 isoforms are indeed the channels responsible for Ca$^{2+}$-activated Cl$^-$ conductance in zebrafish skeletal muscle, we implemented a short interfering (siRNA) knock-down strategy previously described to work well in zebrafish[42,43], since zebrafish ANO1 KO model strains are not available to date. Using the BLOCK-iT™ RNAi Designer algorithm (ThermoFisher Scientific) we designed a series of siRNA expressing DNA constructs targeting the 5′-untranslated region as well as the open reading frame of both ANO1 isoforms. Out of the ten siRNAs expressed in normal zebrafish myotubes seven showed significant ($P < 0.05$) knock-down of Ca$^{2+}$-activated Cl$^-$ currents from 20.6 to 53.1% ($n = 8$–34) compared to a control scrambled siRNA ($n = 28$) (see Method section). The strongest current knock-down ($53.1 \pm 3.6\%$, $n = 34$) was obtained with siRNA targeted against a region, showing 91% homology

between ANO1-a and ANO1-b isoforms (nucleotide positions 2199–2219 and 2073–2093, respectively). In contrast to the highly significant ($P < 0.001$) ANO1 current reduction (Fig. 4a), SR Ca$^{2+}$ release was indistinguishable ($P > 0.05$) from control scrambled siRNA expressing myotubes (Fig. 4b), indicating that the observed ANO1 current reduction cannot be attributed to putative siRNA off-target effects hampering SR Ca$^{2+}$ release. Altogether, our results explicitly ascertain that the channel responsible for zebrafish skeletal muscle Ca$^{2+}$-activated Cl$^-$ conductance is ANO1.

**ANO1 is localised in the sarcolemma.** Since ANO1 is activated by SR Ca$^{2+}$ release, which is initiated by the allosteric interaction between DHPR and RyR1, we next tested if ANO1 channels are located closely to these Ca$^{2+}$ release sites in the triadic junctions or reside in the sarcolemma, or are rather distributed in both the subcellular domains. Immunolocalisation assays on zebrafish skeletal myotubes with anti-ANO1 and anti-DHPRα$_{1S}$ antibodies suggest that ANO1 does not reside in the triads like DHPRα$_{1S}$, but is exclusively expressed in the sarcolemma (Fig. 5a).

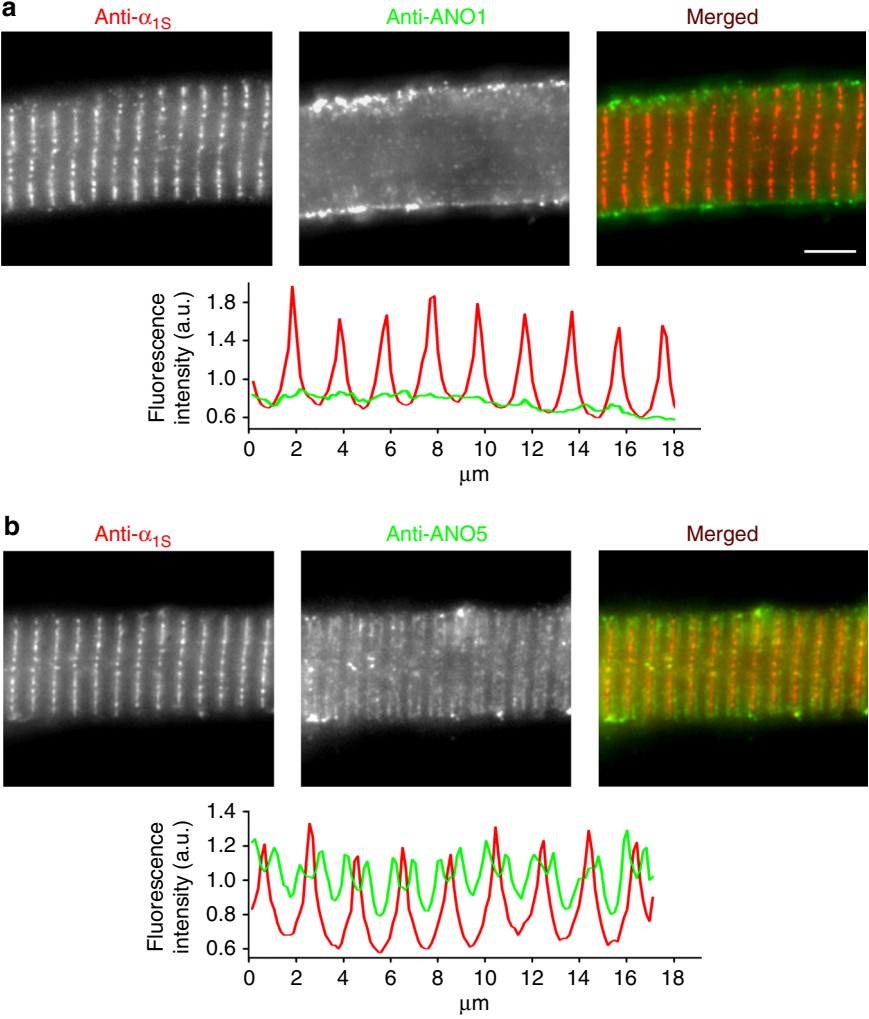

**Fig. 5** ANO1 is localised in the zebrafish skeletal muscle surface membrane. Representative images of normal zebrafish myotubes double immunolabelled for DHPR$\alpha_{1S}$ subunit (anti-$\alpha_{1S}$) and ANO1 (anti-ANO1) or ANO5 (anti-ANO5) isoforms showing their respective subcellular distribution. **a** ANO1 is expressed in the sarcolemma (centre image) and does not co-localise with DHPR$\alpha_{1S}$ (left image) in the triads (merged image). Scale bar, 5 µm. Fluorescence intensity profile (bottom) obtained by measuring the average fluorescence intensity along a horizontal line on the merged image demonstrates regular t-tubular periodicity of the DHPR$\alpha_{1S}$ (red) but a complete lack of a structured ANO1 signal (green). **b** Double immunofluorescence labelling of DHPR$\alpha_{1S}$ (left image) and ANO5 (centre image) under similar conditions clearly revealed anti-ANO5 signal on both sides of the t-tubular membrane (merged image) and thus indicating SR membrane localisation of ANO5. Fluorescence intensity profile displays distinct but regular spatial periodicity with one red peak (DHPR signal) flanked by two green peaks (ANO5 signal)

To test for the accuracy of our fixation procedure and resolution of our immunolocalisation approach for ANO protein detection, we immunostained zebrafish myotubes for another member of the ANO family expected to be present in the skeletal muscles of all vertebrate species, namely ANO5 (Supplementary Fig. 1a). Several lines of evidence suggest that ANO5, expressed in skeletal muscles of human and mouse[44] is essential for the development and maintenance of skeletal muscle[45]. Recessive mutations in ANO5 cause severe myopathies like gnathodiaphyseal dysplasia, GDD[45,46], limb-girdle muscular dystrophy type-2L, LGMD2L[47], and Miyoshi muscular dystrophy-3, MMD3[47,48]. Immunolabeling of zebrafish myotubes with anti-ANO5 antibody showed a similar expression pattern as observed previously for some SR targeted proteins like calsequestrin and the Ca$^{2+}$ sensor D1ER[49–51]. Consequently, we can conclude that ANO5 is expressed in the triadic SR membrane juxtaposing the t-tubules which were immunostained by anti-DHPR$\alpha_{1S}$ antibody (Fig. 5b). The strong and clear signal of ANO5 in zebrafish myotubes suggests that with identical cell preparation, fixation, and

fluorescent staining procedures ANO1 would have certainly been detected in the triads if it would not solely reside in the sarcolemma.

To further confirm the specificity of ANO1 surface expression, we performed immunolocalisation assays on zebrafish skeletal myotubes expressing siRNA 2073–2093 (ANO1-b numbering) which was most effective in knocking-down the ANO1 current (Fig. 4a). Quantification of the sarcolemmal ANO1 fluorescence signal showed a significant ($P < 0.001$) reduction of ANO1 in myotubes expressing the ANO1-specific shRNA construct ($61.4 \pm 1.8\%$, $n = 128$) compared to control myotubes ($100.0 \pm 3.7\%$, $n = 161$) expressing the scrambled shRNA construct (Supplementary Fig. 3). Altogether, results from the above experiments indicate the exclusive sarcolemmal localisation of ANO1 channels in zebrafish skeletal myotubes.

**SR Ca$^{2+}$ release awakes ANO1 outward current close to maximum.** To elucidate if ANO1 currents are activated only

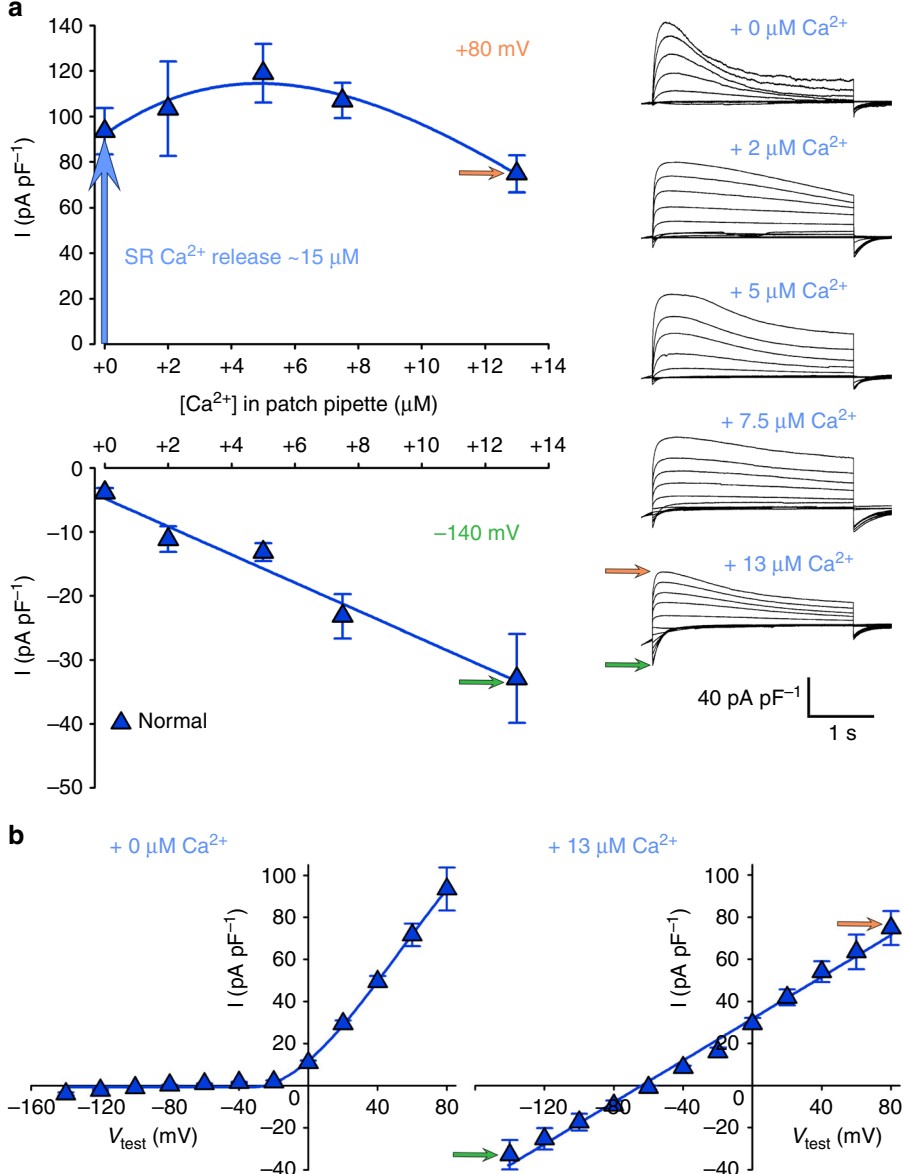

**Fig. 6** ANO1 outward current is activated close to maximum by SR Ca$^{2+}$ release. **a** Ca$^{2+}$-concentration dependence experiments of ANO1 currents were performed by elevating cytosolic Ca$^{2+}$ concentrations via perfusion with the patch pipette solution from the physiological level (+0, $n = 5$) to +2 ($n = 5$), +5 ($n = 9$), +7.5 ($n = 5$), and +13 μM ($n = 8$). Plots of Ca$^{2+}$ dependence of the outward Cl$^-$ current at +80 mV (top graph) and Ca$^{2+}$ dependence of the inward Cl$^-$ current at −140 mV (bottom graph) are shown. Vertical blue arrow indicates Cl$^-$ current stimulation by SR Ca$^{2+}$ release, which raises cytosolic Ca$^{2+}$ to ~ 15 μM[52]. Representative recordings of ANO1 currents (3-s pulse protocol) at different elevations of intracellular Ca$^{2+}$ concentrations are depicted on the right. Scale bars, 1 s (horizontal), 40 pA pF$^{-1}$ (vertical). **b** Current–voltage relationship under physiological conditions i.e., +0 Ca$^{2+}$ addition ($n = 5$, left graph) and after addition of +13 μM Ca$^{2+}$ to the cytosol ($n = 8$, right graph). Orange arrow indicates $I_{max}$ at +80 mV and green arrow at −140 mV. Data are presented as mean ± s.e.m.; $P$ determined by unpaired Student's $t$-test

marginally or to a substantial extent by ~ 15 μM of Ca$^{2+}$ ions released during EC coupling[52], we carried out ANO1 Ca$^{2+}$-dependence experiments by raising the Ca$^{2+}$ concentration in the cytosol via perfusion with the patch pipette solution. With a 3-s depolarisation protocol from −140 to +80 mV in 20-mV increments we observed that at +80 mV under normal SR Ca$^{2+}$ release conditions (i.e., +0 Ca$^{2+}$ addition) the ANO1 current reached 93.47 ± 10.15 pA pF$^{-1}$ ($n = 5$), which is already ~80% of the maximum current yielded after cytosolic addition of +5 μM Ca$^{2+}$ (118.97 ± 12.92 pA pF$^{-1}$, $n = 9$) (Fig. 6a). While the ANO1 outward current, recorded at +80 mV shows a bell shaped Ca$^{2+}$ dependence (Fig. 6a, upper graph), the ANO1 inward current at

−140 mV displays a linear Ca$^{2+}$ concentration–current relationship (Fig. 6a, lower graph). These specific current properties sufficiently explain the ANO1 current–voltage (I–V) relationship during depolarisation-induced SR Ca$^{2+}$ release under physiological conditions (Fig. 6b, left graph). The cytosolic Ca$^{2+}$ concentration at negative potentials, below or at the initiation of SR Ca$^{2+}$ release around −40 mV (Supplementary Fig. 4a) is apparently too low to induce considerable ANO1 currents. Only intracellular addition of micromolar Ca$^{2+}$ can elicit ANO1 inward currents at negative potentials (Fig. 6a, lower graph), which finally yields a linear I–V curve at +13 μM of Ca$^{2+}$ addition (Fig. 6b, right graph).

**Zebrafish skeletal muscle ANO1 channel is a bona fide CaCC.** Similar to mammalian ANO1 channels[24,41], zebrafish skeletal muscle ANO1 is gated synergistically by intracellular $Ca^{2+}$ ions and membrane potential, showing strong outward rectification at low intracellular $Ca^{2+}$ concentrations—a biophysical hallmark of CaCCs. Under physiological conditions (+0 cytosolic $Ca^{2+}$ addition), the skeletal muscle ANO1 I–V curve shows a zone of combined $Ca^{2+}$- and voltage dependence between −40 and +10 mV and a pure voltage dependence zone between +10 and +80 mV, defined by the raising phase or plateau phase of SR $Ca^{2+}$ release, respectively (Supplementary Fig. 4a). Outward rectification was still observed after cytosolic addition of low (+5 µM) $Ca^{2+}$ ions via the patch pipette (Supplementary Fig. 4b). Regardless of whether $Ca^{2+}$ ions were added to the cytosol of normal or *relaxed* myotubes, inward currents were very small compared to the respective outward currents and consequently lead to a pronounced nick in the I–V curves (Supplementary Fig. 4b). As expected, outward currents recorded from normal myotubes were more pronounced compared to *relaxed* myotubes, due to additional current amplification by SR $Ca^{2+}$ release. Moreover, a linear I–V curve at high $[Ca^{2+}]$ (+13 µM $Ca^{2+}$ addition) (Fig. 6b, right graph) depicts the competence of the skeletal muscle ANO1 channel for voltage-independent pore opening i.e., entry into a constitutively active mode.

Altogether, outwardly rectifying as well as ligand/voltage-gating properties of zebrafish skeletal muscle ANO1 channels classify them as bona fide voltage-sensitive $Ca^{2+}$-activated $Cl^-$ channels, analogous to those expressed in mammalian gastrointestinal epithelia or smooth muscles[40].

**Zebrafish skeletal muscle ANO1 coexists with ClC.** Did the evolutionary innovative concept of the $Ca^{2+}$-activated $Cl^-$-channel ANO1 expression in zebrafish skeletal muscle lead to extinction of the solely voltage-gated, canonical skeletal muscle $Cl^-$ channel ClC-1 found in mammals[53–55]? ClC-1 and ClC-2 mRNAs were previously identified in zebrafish skeletal muscle[56] but putative ClC currents remained uncharacterised. To investigate this, we attempted to record ClC currents in zebrafish myotubes using a voltage-clamp protocol previously described for recording ClC-1 currents in mouse FDB fibres[57]. To fully activate ClC currents[57], normal zebrafish myotubes were stimulated with a 250-ms prepulse from a holding potential of −40 to +60 mV (Fig. 7a) and as expected, we observed robust ANO1 currents (Fig. 7b, left trace). During the 500-ms test pulses to positive test potentials, the expected 'contamination' with ANO1 currents could be minimised by starting the recordings with negative test potentials and hence taking advantage of the pronounced ANO1 current rundown as depicted in Fig. 2d. These recording conditions allowed us to measure a current even at +60 mV in normal myotubes (red trace) which mainly reflects the ClC current (Fig. 7b). However, unquestionably non-contaminated ClC recordings could be harvested only from *relaxed* myotubes (Fig. 7c), where ANO1 was not activated due to the lack of SR $Ca^{2+}$ release (Fig. 1c). Consistent with these findings, we observed a dramatic reduction in peak ClC current density (77% at −140 mV and 66% at +60 mV) in the presence of 1 mM 9AC, a blocker of ClC-1 channels[57] (Supplementary Fig. 5).

As shown in Fig. 7d, the zebrafish ClC current displays a linear I–V relationship with an inward component at potentials below −40 mV and an outward current above −40 mV. Notably, in contrast to ANO1, the ClC current does not show outward rectification. ClC currents at potentials above −10 mV are significantly ($P < 0.001$) smaller than the corresponding ANO1 currents and hence might have only a minor contribution to the total skeletal muscle $Cl^-$ influx during membrane depolarisation.

On the contrary, the ClC inward current component at negative potentials, accounting for 80% of the total membrane conductance in resting human muscle and thus ensuring electrical stability, is a characteristic of skeletal muscle ClC[55]. Lastly, this ClC inward current explains slight $Cl^-$ currents that we found in some of our experiments, like in ANO1-current free *relaxed* myotubes (Fig. 2b, d, right panel) or in normal myotubes at −140 mV without cytosolic $Ca^{2+}$ addition (Fig. 6a, lower graph).

**Skeletal muscle ANO1 current accelerates AP repolarisation.** We were interested why the euteleost species zebrafish evolved such pronounced CaCC currents in skeletal muscle—a phenomenon that has not been reported from mammals. Our prime working hypothesis was that this CaCC current, that immediately follows EC coupling-induced SR $Ca^{2+}$ release (Fig. 1a, Supplementary Fig. 4a), plays a role in acceleration of the skeletal muscle action potential (AP) termination. Conventionally, voltage-gated $Na^+$ influx and delayed $K^+$ efflux contribute to the depolarisation and repolarisation phase of an AP, respectively[58]. However, a number of studies in cardiac and neuronal tissues showed that also CaCC currents are involved in shaping of the AP repolarisation phase[59–62]. Application of the CaCC blocker 4,4′-diisothiocyanatostilbene-2,2′-disulphonic acid (DIDS) or reduction of the $Cl^-$ concentration induced significant broadening of cardiac muscle APs in pig ventricular myocytes[59] as well as in rabbit ventricular and atrial myocytes[60,61]. Broadening of AP width was also observed in hippocampal pyramidal neuronal slices upon ANO2 current reduction, either by applying CaCC blockers niflumic acid (NFA) and 5-nitro-2-(3-phenylpropylamino) benzoic acid (NPPB) or by silencing ANO2 expression with short hairpin RNAs[62].

Consequently, to investigate a putative role of the zebrafish skeletal muscle ANO1 in shaping of the AP repolarisation phase, we performed whole-cell current-clamp electrophysiology on *relaxed* and normal myotubes. Interestingly, congruent to our hypothesis, the $AP_{1/2}$ width recorded from *relaxed* myotubes, which lack the SR $Ca^{2+}$ release-activated CaCC current (Figs 1 and 2), was with 2.84 ± 0.28 ms ($n = 13$) nearly double as wide ($P < 0.001$) as recorded from normal myotubes with 1.60 ± 0.14 ms ($n = 14$) (Fig. 8a). However, the AP time to peak was comparable ($P > 0.05$) between *relaxed* and normal myotubes (1.28 ± 0.12 ms, $n = 13$ and 1.28 ± 0.07 ms, $n = 14$, respectively) (Fig. 8a). Finally, to confirm the results obtained from *relaxed* myotubes under standard $Cl^-$ conditions, we performed current-clamp recordings on normal myotubes under $Cl^-$-free conditions (see Methods). The $AP_{1/2}$ width (2.91 ± 0.57 ms, $n = 10$) as well as the AP time to peak (1.17 ± 0.05 ms, $n = 10$) was comparable ($P > 0.05$) between normal myotubes under $Cl^-$ free conditions and *relaxed* myotubes (Fig. 8b). Thus, both series of experiments strongly indicate that the ANO1 $Cl^-$ influx in zebrafish skeletal muscle plays a substantial role in modulating the AP by accelerating the repolarisation phase.

**Only ANO1-current accelerated APs enable proper spike trains.** Apparently, single APs have only limited importance in skeletal muscle functioning. Sustained muscle contractions are evoked only under conditions when motor neurons emit higher frequency APs, resulting in fusion of contractions (tetanic contraction). Muscle contractions start to sum up beyond stimulation frequencies of 5–20 Hz until the response forms a smooth ramped increase of tetanic contraction, which—depending on species and muscle type—reaches 90% of its maximum fusion rate (complete tetanus) at 20–60 Hz. Increased stimulation frequency leads to increased force production and a maximum force of 90% is generated around 60–80 Hz[28–31]. The usual firing rate of

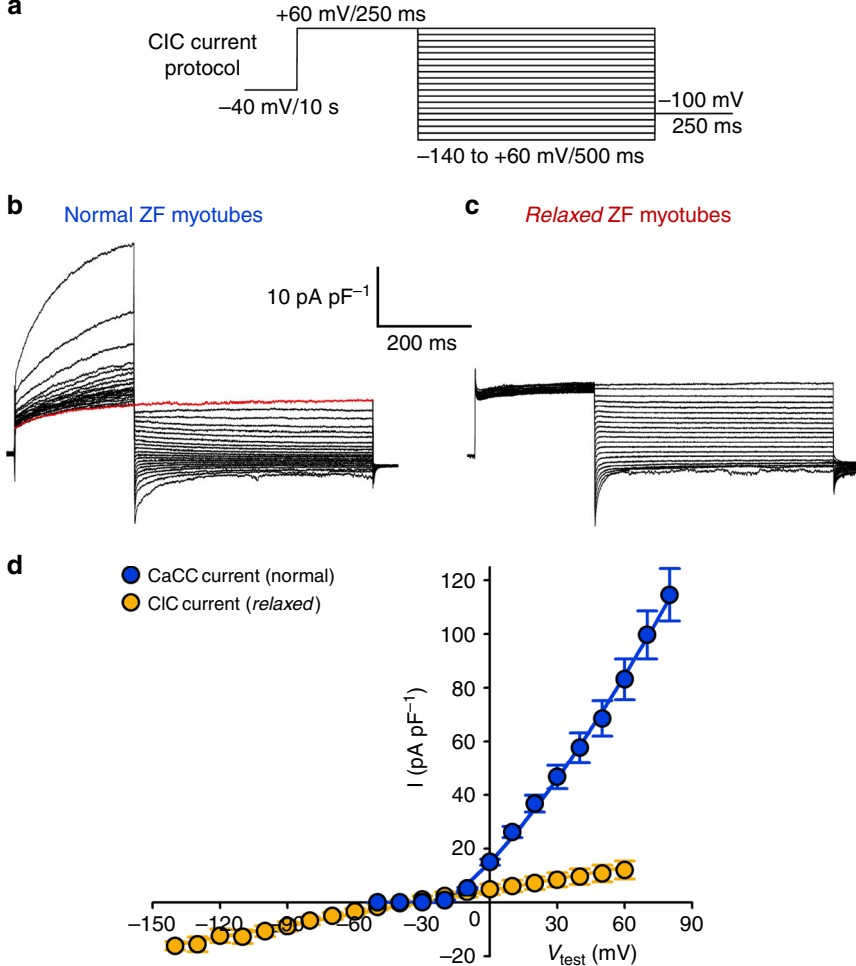

**Fig. 7** Zebrafish skeletal muscle expresses ClC current beside ANO1 current. **a** To elicit ClC currents, a pulse protocol from −140 to +60 mV in 10-mV increments, preceded by a 250-ms prepulse to +60 mV to fully activate ClC currents, as described previously[57] was used. **b** Representative recording from normal myotubes ($n = 8$) demonstrating ClC currents contaminated by ANO1 currents at positive test potentials. Minimisation of contamination in order to harvest close-to-accurate ClC current recordings at positive potentials was achieved by starting the recordings with negative test potentials and thus taking advantage of the ANO1-specific CamKII-induced current rundown. Red trace indicates the last current recorded at +60 mV. Scale bars, 200 ms (horizontal), 10 pA pF$^{-1}$ (vertical). **c** Representative non-contaminated ClC recording from *relaxed* myotubes ($n = 6$), lacking the SR Ca$^{2+}$ release-activated ANO1 currents. **d** Overlay of the clean ClC I–V curve from *relaxed* myotubes ($n = 6$) and the ANO1 I–V curve from normal myotubes ($n = 5$) (from Fig. 1d), displays the significant difference ($P < 0.001$) between ClC and ANO1 (CaCC) Cl$^−$ currents, starting from the test potential ($V_{test}$) of 0 mV. At +60 mV, ClC displays $I = 12.03 \pm 3.38$ pA pF$^{-1}$ while ANO1 reaches $I = 83.12 \pm 7.62$ pA pF$^{-1}$. Data are presented as mean ± s.e.m.; $P$ determined by unpaired Student's $t$-test

vertebrate motor neurons during voluntary muscle contraction is within this tetanic range[32].

To investigate the physiological impact of the skeletal muscle AP width on trains of APs, we performed whole-cell current-clamp recordings in the physiological frequency range from normal and *relaxed* myotubes. Myotubes were subjected to increasing stimulation frequencies between 35 and 80 Hz, in 5-Hz increments to elicit 500-ms long trains of APs with 100-ms recovery intervals. As shown in Fig. 9, there are significant differences ($P < 0.05$) in the kinetics of AP trains at all spike frequencies between normal and *relaxed* myotubes. This is due to piling up (Fig. 9b) of the broader APs in *relaxed* myotubes (Fig. 9d), which leads to the development of a high depolarised plateau[63], evidently because the sarcolemma is unable to adequately repolarise between the broadened APs. The formation of these depolarised plateaus can already be observed at lowest firing rates (Fig. 9a, c). On the contrary, depolarised plateaus—as the manifestation of membrane potential derailment in *relaxed*

myotubes—hardly develop in normal myotubes even at the highest tested AP train frequencies (Fig. 9a, c). These results clearly demonstrate the crucial influence of the ANO1 current on the electrical membrane stability.

Overall, our data show that zebrafish skeletal muscle contains at least two distinct Cl$^−$ channels, namely ANO1 and ClC. While ClC is most likely crucial for its canonical function of stabilising the sarcolemmal resting potential, it seems not to play a major role in shaping the AP because above −10 mV ClC currents are significantly ($P < 0.001$) smaller than ANO1 currents (Fig. 7d) and thus are not sufficient to support ANO1 in accelerating the AP repolarisation phase. Furthermore, this is supported by the striking similarity in AP kinetics (similar AP$_{1/2}$ width) between normal myotubes under Cl$^−$ free recording conditions (where evidently neither ANO1 nor ClC can conduct Cl$^−$ ions) and *relaxed* myotubes under regular Cl$^−$ conditions (where ClC conductance is intact) (Fig. 8b).

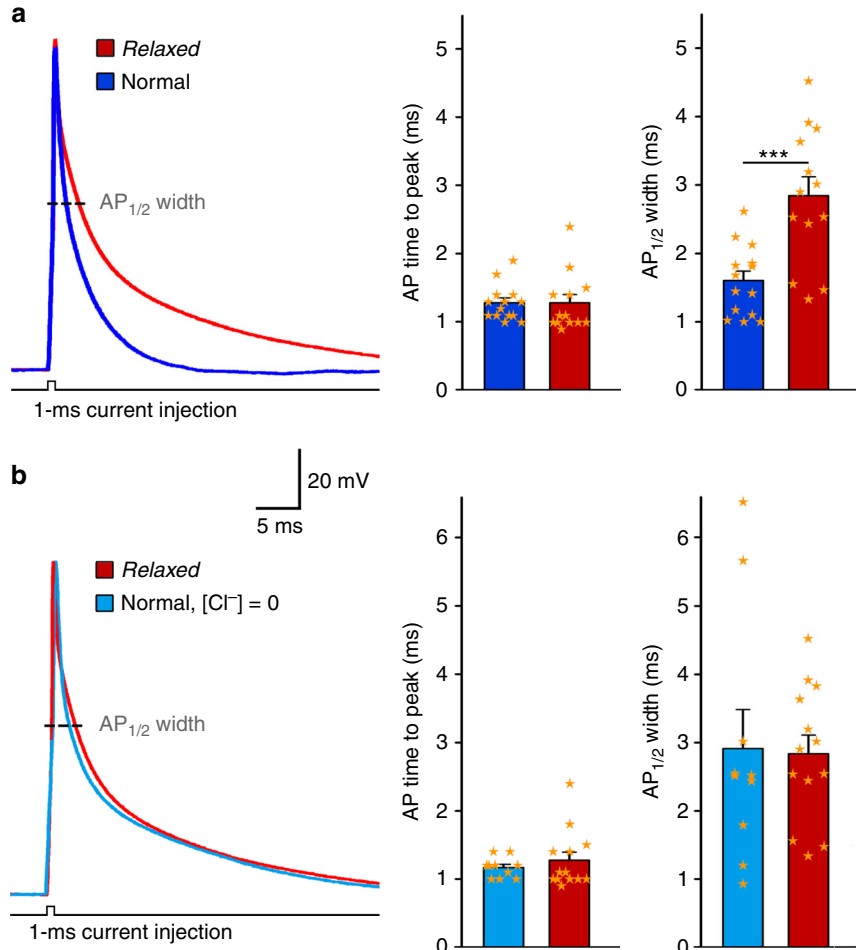

**Fig. 8** The ANO1 current accelerates skeletal muscle AP repolarisation. **a** Overlay of representative AP recordings from normal (dark blue) and *relaxed* (red) myotubes under standard external Cl⁻ conditions (142 mM). Bar graphs depict AP time to peak and AP½ width of normal ($n = 14$) compared to *relaxed* ($n = 13$) myotubes. Scale bars, 5 ms (horizontal), 20 mV (vertical). **b** Overlay of representative AP recordings from normal myotubes in Cl⁻-free external solution ($[Cl^−] = 0$) (light blue) and *relaxed* myotubes in standard external Cl⁻ solution (red). Bar graphs show AP time to peak and AP½ width of normal myotubes in Cl⁻-free external solution ($n = 10$) compared to *relaxed* myotubes under standard Cl⁻ conditions ($n = 13$). Bars represent mean ± s.e.m. and overlaying orange stars indicate individual data points; ***$P < 0.001$ determined by unpaired Student's *t*-test

## Discussion

ANO1, a $Ca^{2+}$-activated $Cl^−$ channel (CaCC) has never been reported to functionally express in skeletal muscle. In this study we identified and characterised the expression pattern, subcellular distribution and physiological role of ANO1 in zebrafish skeletal muscle. Immunocytochemical results provide direct evidence for the sarcolemmal localisation of ANO1 and RT-PCR amplification experiments revealed the expression profile of the two ANO1 isoforms, with ANO1-a in superficial slow/red and ANO1-b in deep fast/white skeletal musculature of zebrafish. Using a broad range of experiments, we characterised ANO1 as a bona fide CaCC which is gated synergistically by membrane potential and intracellular $Ca^{2+}$ concentration and is activated close to maximum by the SR $Ca^{2+}$ release during excitation-contraction coupling. Furthermore, ANO1 coexists with ClC in zebrafish skeletal muscle but only ANO1-current accelerated APs enable proper spike trains crucial for high speed muscle contractions. ClC seems not to play a role in shaping the AP, rather stabilises the sarcolemmal resting potential.

Subsuming these findings, we postulate a model (Fig. 10) where depolarisation-induced SR $Ca^{2+}$ release, activated via the DHPR-RyR1 interaction, leads to more or less in-parallel binding of $Ca^{2+}$ ions to (i) troponin C for initiation of muscle contraction

and (ii) ANO1 for induction of a transient $Cl^−$ influx. This $Ca^{2+}$-release-induced $Cl^−$ influx, together with the delayed $K^+$ efflux via the voltage-gated ($K_V$) channels, causes a quick drop in the membrane potential ($V_m$) during the decline phase of the AP, thereby repolarising the sarcolemma and returning the electrochemical gradient to the resting state. Due to the rapid decline of cytosolic $Ca^{2+}$ by the fast pumping back action of the sarco/endoplasmic reticulum $Ca^{2+}$-ATPase (SERCA)[64], ANO1 is prohibited from $Cl^−$ efflux at negative membrane potentials and thus is strictly outwardly rectifying. The resulting short-term APs, due to the ANO1-$K_V$ synergistic action, are evidently the basis of electrical membrane stability that enables accelerated muscle stimulation rates for high speed (tetanic) muscle contractions with increased force production[28–32]. Altogether, this is crucial for high speed swimming which is vital in the aquatic prey-predator context. Apparently, the phylogenetically highest advanced skeletal muscles of euteleost species[4,5], like zebrafish, developed this innovative 'high-speed gear' by adding ANO1-mediated $Ca^{2+}$ release-induced $Cl^−$ influx to the skeletal muscle EC coupling machinery.

Future investigations on a double transgenic zebrafish strain that selectively expresses fluorescent proteins mCherry and GFP in slow/red and fast/white muscles, respectively, will enable us to

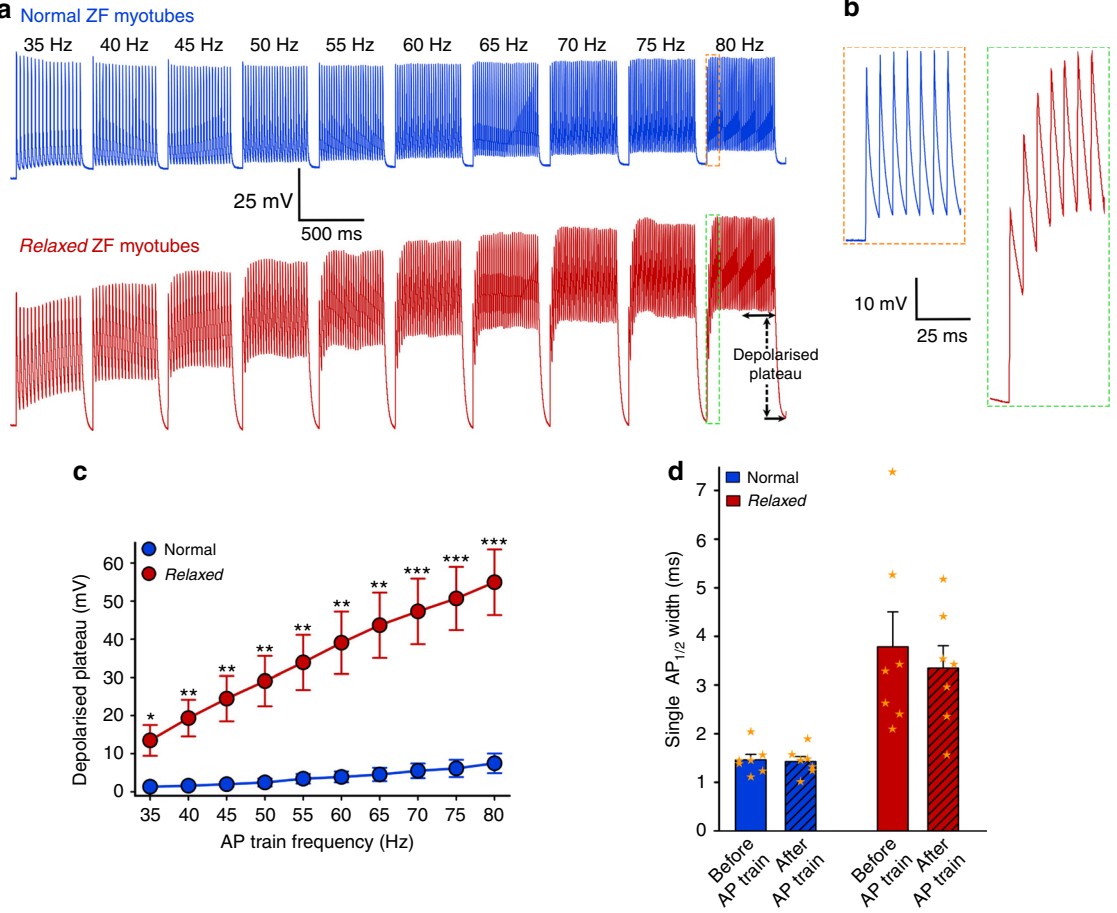

**Fig. 9** The ANO1 current facilitates proper skeletal muscle spike trains. **a** Representative recordings of 500-ms trains of APs from normal (upper traces) and *relaxed* (lower traces) myotubes at increasing stimulations frequencies from 35 to 80 Hz in 5-Hz increments with 100-ms inter-train intervals. *Relaxed* myotubes display pronounced depolarised plateaus in a frequency-dependent mode due to the summation of broader APs. For calculating the size of depolarised plateaus, the second half of the 500-ms AP trains (indicated by a horizontal double-headed arrow) was taken into consideration. Vertical dashed arrow symbolizes the size of the depolarised plateau at 80 Hz. Scale bars, 500 ms (horizontal), 25 mV (vertical). **b** Train of APs on an expanded time base showing piling up of broader APs in *relaxed* myotubes (red trace) compared to normal myotubes (blue trace) during the initial phase of the 80-Hz train. Scale bars, 25 ms (horizontal), 10 mV (vertical). **c** Depolarised plateau sizes were significantly different at all tested frequencies between *relaxed* ($n = 8$) and normal ($n = 7$) myotubes. **d** Mean single AP$_{1/2}$ width before and after the AP train recordings from the same set of normal and *relaxed* myotubes. Mean AP$_{1/2}$ width was indistinguishable ($P > 0.05$) when recorded before or after the trains of APs in normal (1.46 ± 0.11 ms; 1.43 ± 0.10 ms; $n = 7$) as well as in *relaxed* (3.79 ± 0.72 ms; 3.35 ± 0.46 ms; $n = 7$) myotubes. However, the difference in AP$_{1/2}$ width was highly significant ($P < 0.01$) between normal and *relaxed* myotubes, regardless if recorded before or after the trains of APs. Data are presented as mean ± s.e.m. and overlaying orange stars indicate individual data points; *$P < 0.05$, **$P < 0.01$, ***$P < 0.001$ determined by unpaired Student's $t$-test

gain deeper insights into this mechanism of AP acceleration by studying the putatively distinct input of the muscle-type specific (Fig. 3) ANO1 isoforms, identified in this study. According to our model, AP acceleration is expected to be more pronounced in the fast/white musculature which is dedicated to burst activities and thus dependent on high frequency activation.

## Methods

**Zebrafish care.** Care and maintenance of adult zebrafish, wild-type (wt) and heterozygous for the DHPRβ$_1$-null mutation *relaxed* (*red$^{ts25}$*)[27], obtained from the Max Planck Institute (Tübingen, Germany), was according to the established procedures[65,66] and was approved by the Tierethik-Beirat of the Medical University of Innsbruck and Bundesministerium für Wissenschaft, Forschung und Wirtschaft.

**RT-PCR detection.** Total RNA was extracted from superficial slow (red) and deep fast (white) skeletal muscle of multiple adult zebrafish individually, using the RNeasy Mini Kit (Qiagen) and reverse-transcribed using random primers and M-MLV reverse transcriptase (Promega). PCR primer sequences for amplification of DNA fragments from the two isoforms of ANO1 and ANO2, as well as from

DHPRα$_{1S}$-a and α$_{1S}$-b subunits, used as positive controls to determine the purity of the muscle tissue preparations[4], were designed according to the sequences deposited in GenBank database (Supplementary Table 1). Quantification of the band intensities was done using ImageJ (open source). Identity and fidelity of the RT-PCR products was confirmed by sequence analysis (Eurofins, Germany).

**Primary culture of myotubes.** Myoblasts from 2-dpf *relaxed* zebrafish, homozygous or heterozygous for the DHPRβ$_1$-null mutation[27], or wt zebrafish were isolated and cultured as described[67]. Homozygous *relaxed* mutants were identified by their inability to move in response to tactile stimulation and motile 'normal' siblings (heterozygous and wt) were used as controls. Myotubes were cultured for 4–6 days in a humidified 28.5 °C incubator in L-15 medium supplemented with 3% foetal calf serum, 3% horse serum, 4 mM L-glutamine and 4 U/ml penicillin/streptomycin.

**siRNA knock-down.** Short hairpin RNA (shRNA) constructs that are intracellularly processed into short interfering RNAs (siRNAs), making them putatively suitable to inhibit gene expression, were designed according to described procedures for zebrafish[42,43]. 21 nucleotides (nt) long targeting sequences against ANO1-a and ANO1-b RNA were designed using BLOCK-iT™ RNAi Designer software (https://rnaidesigner.thermofisher.com/rnaiexpress/) and only hits with

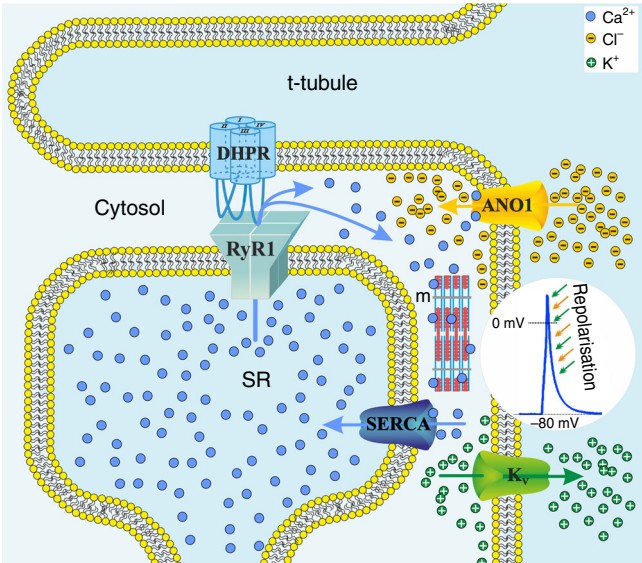

**Fig. 10** Model of ANO1 activation by SR Ca$^{2+}$ release as the basis for acceleration of AP repolarisation. Schematic representation of the skeletal muscle triad with sarcolemmal t-tubular invagination (t-tubule) adjacent to the sarcoplasmic reticulum Ca$^{2+}$ store (SR) and membrane localisation of some selected channels and pumps involved in cytosolic (Cytosol) Ca$^{2+}$ handling. Initially, depolarisation-induced conformational changes in the DHPR are transmitted to the RyR1, which leads to pore opening and release of Ca$^{2+}$ ions (blue spheres) from the SR stores. Cytosolic Ca$^{2+}$ activates contraction of muscle fibres (m) and also binds to intracellular ANO1 Ca$^{2+}$ binding sites[19], to activate massive Cl$^-$ influx (yellow spheres) into the cytosol. Cl$^-$ influx together with simultaneous K$^+$ efflux (green spheres) via the voltage-gated K$^+$ channels (K$_V$), synergistically and rapidly reduces the membrane potential to accelerate AP repolarisation (inset, green and yellow arrows). Finally, the SERCA pumps back Ca$^{2+}$ into the SR and thus resetting the system

the highest score (5 stars) were considered. The siRNA sequences used were as follows: −156 to −136, 1725–1745 (ANO1-a nt numbering); 653–673, 914–934, 1434–1454, 1496–1516, 1593–1613, 2073–2093, 2271–2291, and 2327–2347 (ANO1-b nt numbering). Knock-down effects of ANO1-targeting sequences were compared to a scrambled control sequence (TCACAATAGTACCAAGCATGA) that showed no homology when blasted to the zebrafish genome (https://blast.ncbi.nlm.nih.gov/Blast.cgi). For construction of a plasmid apt for expressing siRNAs under the control of a CMV promoter, the SalI-NotI DNA fragment containing the original zebrafish miR30 hairpin region from plasmid pmE-actin-exon+intron-miR30-CFP[43] was inserted into the corresponding polylinker sites of expression vector pCI-neo (Promega). Following deletion of the BbsI RE site in the pCI-neo backbone by replacing nts G957/A958 with TT via fusion PCR, the ANO1 or control scrambled siRNA target sequences could be inserted into the corresponding BbsI sites of the miR30 hairpin region. Sense and antisense 71-mer shRNA-coding oligonucleotides in a typical passenger-loop-guide assembly[42,43] were synthesised and annealed at a decreasing temperature gradient. Integrity of all DNA constructs was confirmed by sequence analysis (Eurofins, Germany). 1-dpf wt zebrafish myoblasts were transfected with shRNA constructs using AMAXA$^{TM}$ nucleofector kit[34] and subsequently the cultured myotubes were used for immunostaining and electrophysiological recordings[34,67].

**Immunostaining.** Six days old cultured normal myotubes were washed with 1x PBS, fixed in 4% paraformaldehyde both supplemented with 100 μM N-benzyl-p-toluene sulphonamide (BTS) and blocked by incubating in 5% goat or rabbit serum, as described in detail[34]. Primary and secondary antibodies, diluted in PBS supplemented with 0.2% BSA and Triton X-100 were as follows: mouse monoclonal antibody 1A against DHPRα$_{1S}$ (1:2,000, MA3-920, Affinity Bioreagents), rabbit polyclonal anti-ANO1 (1:50, ab84115, Abcam), goat polyclonal anti-ANO5 (1:500, sc-169628, Santa Cruz), rabbit anti-GFP (1:5,000, A11122, Invitrogen), secondary goat anti-mouse Alexa Fluor 594, goat anti-rabbit Alexa Fluor 488 (both 1:4,000, A11032 and A11034, respectively, Invitrogen), rabbit anti-goat Cy3 and rabbit anti-mouse FITC (1:500, C2821, and 1:2,000, F9137, respectively, Sigma).

Images were recorded with a cooled CCD camera (Diagnostic Instruments) and MetaVue image processing software (v 6.2, Universal Imaging, PA). Quantification of ANO1 surface membrane expression after siRNA knock-down was determined by measuring the average fluorescence intensity along the periphery of CFP positive myotubes, obtained from at least two different cultures using the MetaVue software.

**Voltage-clamp electrophysiology.** CaCC currents were recorded from cultured myotubes simultaneously with intracellular SR Ca$^{2+}$ release by using 0.2 mM Fluo-4 in the patch pipette (internal) solution, and were evoked by a 200-ms pulse protocol from +80 to −50 mV in 10-mV steps from a holding potential of −80 mV, as described[67]. BTS Myosin-II blocker (100 μM) was continuously present in the bath (external) solution. The standard external solution, with a total Cl$^-$ concentration of 165 mM, contained (in mM): 10 CaCl$_2$, 145 TEA-Cl and 10 HEPES, pH 7.4 with TEA-OH. For Cl$^-$ free external solution, 10 mM CaCl$_2$ and 145 mM TEA-Cl were replaced by 10 mM Ca(OH)$_2$ and 100 mM Aspartate, respectively. The standard Cl$^-$ containing internal solution used was as follows (in mM): 145 Cs-aspartate, 2 MgCl$_2$, 10 HEPES, 0.1 Cs$_2$-EGTA, and 2 Mg-ATP (pH 7.4 with CsOH). The Cl$^-$ free internal solution contained (in mM): 100 Cs-aspartate, 10 HEPES, 0.5 Cs$_2$-EGTA, and 3 Mg-ATP (pH 7.4 with CsOH).

To test the integrity of CaCC in the DHPRβ$_1$-null *relaxed* zebrafish, recordings of CaCC currents following caffeine-induced SR Ca$^{2+}$ store depletion were performed. From a holding potential of −80 mV, 2-dpf myotubes were voltage-clamped at +40 mV for 100 ms followed by a 100-ms step to −120 mV (Fig. 2a). The entire protocol was repeated 100 times and between the 15th and 45th sweep the myotubes were perfused with the standard bath solution supplemented with 8 mM of the RyR agonist caffeine (Merck). Changes in CaCC currents at +40 mV (outward current) and at −120 mV (inward current) before and after application of caffeine were analysed.

To investigate the Ca$^{2+}$-dependence of CaCC currents, intracellular Ca$^{2+}$ concentrations were altered via the patch pipette solution. Using the MaxChelator simulation program (http://maxchelator.stanford.edu), pipette solutions for free Ca$^{2+}$ concentrations ([Ca$^{2+}$]) of 0, 2, 5, 7.5 and 13 μM were adjusted (Supplementary Table 2). Due to the unknown kinetics of the CaCC currents under different Ca$^{2+}$ concentrations contributed by the patch pipette solution in addition to the normal SR Ca$^{2+}$ release, a prolonged 3-s pulse protocol from +80 to −140 mV in 20-mV steps was used.

ClC currents were recorded by applying the voltage-step protocol described by[57] and depicted in Fig. 7a, using standard Cl$^-$-containing external and internal solutions. To confirm that the Cl$^-$ current recordings from *relaxed* myotubes are indeed currents through the skeletal muscle ClC channel, 1 mM 9-anthracene carboxylic acid (9AC), a blocker of ClC-1 channels[57] was added to the external recording solution.

All recordings were performed at room temperature in the whole-cell configuration using the Axopatch 200B amplifier (Axon Instruments Inc., CA), filtered at 1 kHz and digitised at 5 kHz. Data were analysed using ClampFit (v10.0; Axon Instruments) and SigmaPlot (v10.0; Systat Software, Inc.).

**Current-clamp electrophysiology.** Action potentials (APs) were recorded from single myotubes in a perforated patch configuration by using 120 μg/ml of amphotericin B in the patch pipette solution under standard Cl$^-$ and Cl$^-$-free conditions. The standard Cl$^-$-containing bath solution used was as follows (in mM): 130 NaCl, 4 KCl, 2 MgCl$_2$, 2 CaCl$_2$, 10 HEPES, and 10 glucose (pH 7.4 with NaOH). For Cl$^-$-free bath solution NaCl, KCl, MgCl$_2$ and CaCl$_2$ were replaced by Na-aspartate, K-aspartate, Mg-aspartate and Ca(OH)$_2$, respectively. The standard Cl$^-$ containing internal solution consisted of (in mM): 135 K-aspartate, 8 NaCl, 2 MgCl$_2$, 20 HEPES, and 5 EGTA (pH 7.4 with NaOH). For Cl$^-$ free internal solution NaCl and MgCl$_2$ were replaced by Na-aspartate and Mg-aspartate, respectively. Recordings were initiated after amphotericin B lowered the access resistance below 15 MΩ. A small hyperpolarising current was injected to set the membrane potential to −80 mV and APs were elicited by injecting a current of 3.8 nA for 1 ms. Identical current injections were applied to record 500-ms trains of APs from 35 to 80 Hz, in 5-Hz increments bracketed by 100-ms recovery intervals. Data were acquired with an EPC10 amplifier (HEKA Elektronik, Germany) at a sampling rate of 30 kHz, low-pass-filtered at 3 kHz and analysed using FitMaster (v2x73.2, HEKA Elektronik, Germany) and SigmaPlot 10.0 (Systat Software, Inc.).

**General experimental design and statistical analyses.** Sample sizes of zebrafish myotubes or tissues are based on previous publications[4,5,27,34,67], hence, power calculations were not necessary. For siRNA transfection or drug administration assays, myotube cultures according to genotype were arbitrarily allocated to the specific sample groups without the use of an explicit randomisation procedure. Experiments did not require blinding and thus were not performed under blinded conditions. In our study, no data points or samples were excluded from analysis. Based on our previous publications[4,5,27,30,33,34,67] all statistical analyses are considered appropriate. *n*-values represent the number of independent experiments on zebrafish myotubes or tissues, as specified. Variance is similar between comparison groups. All results are expressed as means ± s.e.m. Statistical significance was determined by using unpaired Student's *t*-test. $P < 0.05$ was considered statistically significant and * indicates $P < 0.05$, **$P < 0.01$, and ***$P < 0.001$.

**Reporting summary**. Further information on experimental design is available in the Nature Research Reporting Summary linked to this article.

## Data availability

Data supporting the findings of this study are available from the corresponding authors upon reasonable request.

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

## Acknowledgements
We thank Dr Petronel Tuluc for support in establishing current-clamp recordings and analysis, Dr Hazel Sive for the miR30 backbone clone, Birgit Kagerbauer and Robert B. Janssen for excellent technical assistance. This study was supported by the Austrian Science Fund (FWF) Grant (P23229-B09) and graduate program (W1101-B12) to M.G. and P27392-B21 to M.G. and A.D.

## Author contributions
A.D. and M.G. designed research; S.F.N., A.D. and M.G. performed experiments; A.D. and M.G. wrote the manuscript.

## Additional information

**Competing interests:** The authors declare no competing interests.

