## [Peer Review File · Nature Communications]

Reviewers' Comments:

Reviewer #1:

Remarks to the Author:

Dayal and colleagues have identified a TMEM/ANO family Ca²⁺-activated Cl⁻ channel expressed in zebrafish skeletal muscle that influences repetitive activity by accelerating the repolarization phase of action potentials. This work is potentially impactful because it helps explain a fundamental element of teleosts swimming ability and may provide comparative information regarding mammalian muscle function.

The authors observed a large Cl⁻ conductance in "normal" zebrafish myotubes which was confirmed in Cl⁻ removal experiments. The presence of the Cl⁻ conductance was not dependent on Ca²⁺ flux via L-type Ca²⁺ channels because zebrafish CaV1.1 channels do not conduct Ca²⁺. However, the Cl⁻ conductance appeared to be dependent on CaV1.1-dependent activation of SR Ca²⁺ flux via RyR1, as it was absent in relaxed myotubes null for the excitation-contraction coupling-essential β 1 subunit. Likewise, the Cl⁻ conductance was rescued independently of CaV1.1 activity by application of the RyR agonist caffeine. PCR revealed the Cl⁻ channel as ANO1a/b, a TMEM family channel known to be enriched in the gut but previously not thought to be expressed in skeletal muscle. Immunohistochemistry showed that ANO1 localizes to muscle surface membrane and not the t-tubule network. Since ANO1 channels have an I-V relationship that is distinct from ClC-1 Cl⁻ channels both in magnitude (i.e., bigger) and voltage-dependence (i.e., outwardly rectifying in the presence of elevated intracellular Ca²⁺ levels), the authors posited that the ANO1-mediated Cl⁻ conductance serves to promote repolarization during repetitive muscle activity. Indeed, AP repolarization is much faster in normal myotubes compared to that in relaxed myotubes in which the Ca²⁺-activated Cl⁻ conductance is absent. The repolarization impairment in relaxed myotubes was also evident in elevated plateau potentials at all frequencies tested in current clamp experiments.

Very rigorous, stats OK.

The manuscript is an elegant work with few weaknesses, but the contribution of ANO1 to skeletal muscle physiology in teleost fish still needs to be established on the molecular level---experiments with either siRNA directed to a common sequence in ANO1a/b or ANO1a/b double KO myotubes are necessary to confirm that ANO1 is in fact the channel responsible for the Cl⁻ conductance and for acceleration of AP repolarization.

An image of a myotube in which ANO1a/b expression has been ablated would bolster the IHC data asserting that it resides in the sarcolemma, not in the t-tubules. Alternatively, an image of an XFP-tagged ANO1a/b might be useful in this regard.

The authors touch on the hot button topic of ClC-1 distribution in skeletal muscle on pg. 7, ln. 155. My advice would be to avoid this dispute by deleting this phrase, as it draws the focus away from the current work. The Vergara paper (DiFranco et al., JGP, 2011) probably should be cited here in lieu of the Lamb commentary, if the authors decide to keep as is.

P2. Ln. 50—positions is misspelled.

P10. Ln. 209—myotubes should be FDB fibers.

Reviewer #2:

Remarks to the Author:

The authors report that ANO1, a calcium-activated chloride channel that has never been reported in skeletal muscle, is uniquely expressed in skeletal musculature of zebrafish where it regulates

action potential duration and frequency during muscle contractions. This finding could be of general interest because of many reports on ANO5 in skeletal muscle dysfunction in muscular dystrophy and amyloidosis, but little is known of ANO1. On the other hand, the role ANO1 in skeletal muscle appears to be restricted to zebrafish and thus more of a particularity than a general principle.

The authors detected ANO1a transcripts in red & ANO1b in white muscle, but not ANO2a&b. They should try other probes before excluding ANO2. They should also show western blots (as a supplementary figure) for the ANO1 and 5 antibodies to confirm their specificity as this is their first use. They could also try an ANO2 antibody to confirm lack of staining.

The authors demonstrated a Cl⁻ current activated by SR Ca release that depends on external Cl⁻. This was not present in the DHPR β 1-null zebrafish mutant relaxed and was restored by SR Ca depletion with caffeine. They also describe voltage and Ca dependence of presumed ANO1 current. While this looks and feels like ANO1, ultimately the authors should use CRISPR editing to knockout ANO1 a or b to directly demonstrate their roles.

The muscle action potential was longer in relaxed mutants and with Cl⁻=0, suggesting a role for ANO1 in regulating spike width in zebrafish skeletal muscle, as it does in other cell types. But the currents in Fig 7b are brief – is this a different time scale than 7a? The authors show build up of depolarization in relaxed mutants but not in normal muscle. Is this seen in normal muscle with Cl⁻=0? Of greater physiological relevance, does this occur upon motoneuron. Finally, on a technical point, the mutant shows gradual depolarization and not a 'plateau' potential as the latter is a specific term eg for sustained depolarization in the absence of stimuli that is eliminated by strong, brief hyperpolarization, among other features.

Reviewer #3:

Remarks to the Author:

The manuscript by Dayal, Ng and Grabner employs a sophisticated combination of molecular, immunocytochemical and biophysical approaches to characterize calcium-activated chloride channel expression, subcellular localization and function in zebrafish skeletal muscle.

Using isoform-specific RT-PCR primers, the authors show that ANO1-a is predominantly expressed in red muscle, while ANO1-b is predominantly expressed in white muscle. Immunocytochemical studies indicate that ANO1 channels primarily localize to the surface membrane and not in the transverse tubule membrane. Whole cell patch clamp studies in myotubes from normal and relaxed zebrafish (which lack depolarization-induced elevations in intracellular calcium) provide convincing evidence for a robust chloride conductance consistent with ANO1 activity that requires an increase in cytosolic calcium. Specifically, the authors show the current is abolished by removing extracellular chloride and is absent in myotubes from relaxed zebrafish. Importantly, robust chloride currents could still be elicited in relaxed myotubes during application of 8 mM caffeine to release calcium from intracellular stores (though calcium levels were not measured during these experiments). By dialyzing myotubes with known concentrations of free calcium, the authors show that the chloride current exhibits outward rectification at low calcium levels and linear voltage-dependence at cytosolic high calcium levels (13 μ M), consistent with prior measurements of ANO1 chloride currents. Current clamp studies in myotubes from normal and relaxed zebrafish revealed that the calcium-activated chloride current significantly reduces action potential duration, such that trains of action potentials with a stable negative plateau potential are possible up to frequencies as high as 80 Hz. Thus, while chloride channel function plays a similar role in maintaining normal muscle electrical excitability in both zebrafish and mammals, this is accomplished through two entirely different channels (ANO1 vs CLC1) with the relevant chloride conductances operating across vastly different voltage ranges (depolarized vs hyperpolarized potentials).

This study provides compelling evidence for a robust calcium-dependent chloride conductance in zebrafish skeletal muscle required to maintain electrical stability during high frequency tetanic contractions that is likely crucial for aquatic predator-prey swimming behaviors. In addition, the study provides convincing evidence for another intriguing evolutionary/physiological difference between euteleost and mammalian skeletal muscle. The experiments are carefully performed, appropriately analyzed and the manuscript is well-written.

I only have several relatively minor suggestions, which should help to further strengthen the conclusions and overall impact of this study.

1) Together, the ANO1 mRNA expression and biophysical properties of calcium-activated chloride channel function in zebrafish muscle are strongly suggestive that the currents measured are due to ANO1 channels. However, the results are correlative and definitive data linking the two are lacking. The final nail in the coffin would be to show that the chloride currents are abolished by morpholino-induced ANO1 knockdown. Morpholino ANO1 knockdown studies would also enable addressing another concern regarding the specificity of the antibody used in the immunocytochemical analysis of ANO1 subcellular localization. Fig. 4 presents a single representative image showing ANO1 localization to the surface membrane. Does this antibody only recognize ANO1 in zebrafish muscle? Does a western blot of zebrafish muscle using this antibody show only one band at the expected molecular weight for ANO1 or are multiple proteins recognized by this antibody? Is a similar pattern shown using multiple different ANO1 antibodies? All of these questions could be addressed by showing that the immunofluorescence signal in the surface membrane is abolished following morpholino-induced ANO1 knockdown.

2) From the results in Fig. 6 (particularly those from relaxed myotubes), the authors conclude that a low level of CLC1 current is present in zebrafish muscle. However, this current exhibits a linear current-voltage relationship rather than the prominent inward rectification that is emblematic of CLC1 currents. In order to support the authors' conclusion that this current is indeed due to CLC1 channels, the authors should demonstrate that the current is blocked by a CLC1 channel inhibitor (e.g. 9-anthracene carboxylic acid) and/or morpholino-induced knockdown of CLC1.

3) The recordings of action potential trains in normal and relaxed myotubes in Fig. 8 are elegant. The authors' conclusion that the differences observed in action potential plateau levels are vital for maintaining high speed contractile behavior would be strengthened by showing enhanced or more sustained calcium transient amplitudes during repetitive 80 Hz stimulation trains in myotubes from normal fish compared to that obtained from relaxed fish.

4) Page 3, line 57. In smooth muscle cells, the intracellular chloride concentration is not higher than that of the extracellular space (e.g. see Table 1 in review of Chipperfield and Harper, 2000).

5) Given the results in Fig. 3f (DHPR data shows 24% contamination of red muscle while ANO1-b is 45% in red muscle), it is too strong to state in the results section that ANO1-a is "exclusively" expressed in red muscle (and ANO1-b in white muscle). In the absence of more definitive data, it would be better to stick with the terminology used in the legend to Figure 3 ("predominantly" or "mainly").

6) Page 12, line 261. While 50 Hz is nearly sufficient for tetanus in soleus, a much higher stimulus frequency (125-150 Hz) is actually required to achieve tetanus in fast twitch skeletal muscle.

7) Reference 34. The last name of the first author of the intended reference is Davis, not Alison.

8) The authors go overboard touting the purported phylogenetic superiority of euteleost fish. There are at least 7 or 8 statements similar to "euteleost fishes is evolutionary highly advanced compared to mammals." The statements are made so often that it becomes "preachy." It should

be sufficient to make this assertion once or twice at key places in the manuscript.

Our point-by-point response to the reviewers:

We would like to thank the reviewers for their overall positive judgement of our study and helpful and constructive comments which will certainly improve our manuscript. Accordingly, we have revised the manuscript and implemented suggested changes and additions. All changes in the manuscript text file are highlighted in blue. Detailed answers to the reviewers' comments are as follows:

Our answers to the comments of **Reviewer #1**:

Dayal and colleagues have identified a TMEM/ANO family Ca²⁺-activated Cl⁻ channel expressed in zebrafish skeletal muscle that influences repetitive activity by accelerating the repolarization phase of action potentials. This work is potentially impactful because it helps explain a fundamental element of teleosts swimming ability and may provide comparative information regarding mammalian muscle function.

The authors observed a large Cl⁻ conductance in “normal” zebrafish myotubes which was confirmed in Cl⁻ removal experiments. The presence of the Cl⁻ conductance was not dependent on Ca²⁺ flux via L-type Ca²⁺ channels because zebrafish CaV1.1 channels do not conduct Ca²⁺. However, the Cl⁻ conductance appeared to be dependent on CaV1.1-dependent activation of SR Ca²⁺ flux via RyR1, as it was absent in relaxed myotubes null for the excitation-contraction coupling-essential β 1 subunit. Likewise, the Cl⁻ conductance was rescued independently of CaV1.1 activity by application of the RyR agonist caffeine. PCR revealed the Cl⁻ channel as ANO1a/b, a TMEM family channel known to be enriched in the gut but previously not thought to be expressed in skeletal muscle.

Immunohistochemistry showed that ANO1 localizes to muscle surface membrane and not the t-tubule network. Since ANO1 channels have an I-V relationship that is distinct from ClC-1 Cl⁻ channels both in magnitude (i.e., bigger) and voltage-dependence (i.e., outwardly rectifying in the presence of elevated intracellular Ca²⁺ levels), the authors posited that the ANO1-mediated Cl⁻ conductance serves to promote repolarization during repetitive muscle activity. Indeed, AP repolarization is much faster in normal myotubes compared to that in relaxed myotubes in which the Ca²⁺-activated Cl⁻ conductance is absent. The repolarization impairment in relaxed myotubes was also evident in elevated plateau potentials at all frequencies tested in current clamp experiments.

Very rigorous, stats OK.

The manuscript is an elegant work with few weaknesses, but the contribution of ANO1 to skeletal muscle physiology in teleost fish still needs to be established on the molecular level---experiments with either siRNA directed to a common sequence in ANO1a/b or ANO1a/b double KO myotubes are

necessary to confirm that ANO1 is in fact the channel responsible for the Cl⁻ conductance and for acceleration of AP repolarization.

We agree with the reviewer that the final confirmation for the contribution of ANO1 channels in zebrafish skeletal muscle physiology is the abolishment of this specific Ca²⁺-activated Cl⁻ current by ANO1 knock out or knock down. Since zebrafish ANO1 KO model strains were not available, we decided to use a siRNA knock down strategy as suggested by the reviewer. We designed a series of short hairpin RNA (shRNA) constructs, targeted against both ANO1 isoforms, using the technique previously described to work well in zebrafish (*Dong et al., PLOS ONE, 2009; De Rienzo et al., Zebrafish, 2012*). From the 10 siRNAs that were expressed in normal zebrafish myotubes and tested in patch-clamp experiments, 7 showed significant reduction in Ca²⁺-activated Cl⁻ currents compared to control scrambled siRNA. These experiments confirm that ANO1 is in fact the channel responsible for zebrafish skeletal muscle Ca²⁺-activated Cl⁻ conductance and consequently, for the acceleration of AP repolarization.

In the revised manuscript, siRNA knock-down experiments are exemplified in Figure 4 as reduced CaCC currents and unaltered SR Ca²⁺ release following expression of the most actively blocking siRNA and explicated in the Methods (lines 348 - 368) as well as in the Results section (lines 148 - 162).

An image of a myotube in which ANO1a/b expression has been ablated would bolster the IHC data asserting that it resides in the sarcolemma, not in the t-tubules. Alternatively, an image of an XFP-tagged ANO1a/b might be useful in this regard.

As suggested by the reviewer we tried to strengthen our former ICC data by performing immunofluorescence assays on siRNA expressing myotubes to evaluate the ablation / reduction of ANO1-a/b sarcolemmal expression. And indeed, the sarcolemmal ANO1 signal is significantly reduced in myotubes expressing the most active siRNA (see response above) compared to control myotubes expressing the scrambled siRNA. Thus, consistent with our previous findings (Figure 5a) these new siRNA knock down experiments confirm the sarcolemmal localization of ANO1. Results are demonstrated in the revised manuscript as Supplementary Figure 3 and described in the Methods (lines 378 - 381) and Results sections (lines 182 - 189).

The authors touch on the hot button topic of ClC-1 distribution in skeletal muscle on pg. 7, ln. 155. My advice would be to avoid this dispute by deleting this phrase, as it draws the focus away from the current work. The Vergara paper (DiFranco et al., JGP, 2011) probably should be cited here in lieu of the Lamb commentary, if the authors decide to keep as is.

The reviewer is correct that mentioning the dispute about the CIC-1 distribution in skeletal muscle in this special context might draw the focus away from the current work on ANO1 muscle cell distribution and function. Hence, in the revised manuscript we deleted this phrase and replaced it by the following sentence (lines 169 - 172): “*To test for the accuracy of our fixation procedure and resolution of our immunolocalisation approach for ANO protein detection, we immunostained zebrafish myotubes for another member of the ANO family expected to be present in the skeletal muscles of all vertebrate species, namely ANO5 (Supplementary Fig. 1a).*” Consequently, the Lamb commentary was removed and *Lueck et al., 2010* is now cited as ref. #57.

P2. Ln. 50—positions is misspelled.

This sentence that contained the misspelled word was entirely removed (revised manuscript; line 50). For the rationale please see the comment of reviewer #3 (point 8) and our corresponding answer.

P10. Ln. 209—myotubes should be FDB fibers.

The term “*myotubes*” was corrected to “*FDB fibres*” (revised manuscript; line 229).

Our answers to the comments of **Reviewer #2**:

The authors report that ANO1, a calcium-activated chloride channel that has never been reported in skeletal muscle, is uniquely expressed in skeletal musculature of zebrafish where it regulates action potential duration and frequency during muscle contractions. This finding could be of general interest because of many reports on ANO5 in skeletal muscle dysfunction in muscular dystrophy and amyloidosis, but little is known of ANO1. On the other hand, the role ANO1 in skeletal muscle appears to be restricted to zebrafish and thus more of a particularity than a general principle.

The authors detected ANO1a transcripts in red & ANO1b in white muscle, but not ANO2a&b. They should try other probes before excluding ANO2.

In agreement with the reviewer we used two additional, degenerate (pan) RT-PCR primers to simultaneously amplify ANO2-a and ANO2-b cDNA. RT-PCR amplification from first strands of 3 adult normal whole zebrafish showed robust amplification products with both ANO2 pan-primer pairs, but in contrast, no signal from skeletal-muscle derived first strands from another set of 3 adult normal zebrafish with the same ANO2 pan-primers and under identical PCR conditions. However, control experiment with the same skeletal-muscle derived first strands and a stoichiometric mix of ANO1-a and ANO1-b primers showed strong ANO1-

a/b signals. These new results are depicted in Supplementary Figure 2 of the revised manuscript.

From all these observations we can conclude that ANO2 is not expressed in zebrafish skeletal muscle and articulated this in a sentence in the Results section of the revised manuscript (lines 139 - 141): “*Furthermore, RT-PCR amplification by using two additional ANO2 pan-primer pairs confirmed the non-existence of ANO2 isoforms in skeletal muscle (Supplementary Fig. 2).*”

They should also show western blots (as a supplementary figure) for the ANO1 and 5 antibodies to confirm their specificity as this is their first use. They could also try an ANO2 antibody to confirm lack of staining.

In order to address this query, we would like to take up comment 1 of reviewer #3, stating that all the questions about specificity of the ANO1 antibody “... *could be addressed by showing that the immunofluorescence signal in the surface membrane is abolished following ANO1 knockdown.*” In the revised manuscript we clearly show a significant reduction of the immunofluorescence signal in the surface membrane of myotubes following siRNA-induced ANO1 knock down (demonstrated in Supplementary Figure 3 and discussed in the Results section (lines 182 - 189)). Thus, the specificity of the ANO1 antibody can be confirmed with this ANO1-specific siRNA knock down experiments and as elegantly exemplified by reviewer #3, would indeed answer in advance all these kind of theoretical questions.

Regarding the note that this is the first use of the ANO1 and ANO5 antibodies: Both antibodies were already used previously, like ANO1 in *Crutzen et al., Pflügers Arch, 2016*; *Liu et al., British J. Pharm., 2016*, and ANO5 in *Song et al., BBRC, 2014*; *Hy et al., Adv. Pharmacoevidemiol. Drug Saf., 2015*; *Tian et al., JSM Regen. Med. Bio. Eng., 2015*. To indicate this better we inserted the precise designation of both antibodies in the Methods section of the revised manuscript (lines 373 - 374).

To the last suggestion that we could also try an ANO2 antibody to confirm lack of staining: From our extensive PCR studies (illustrated in Figure 3 and Supplementary Figure 2 of the revised manuscript) we are fully convinced that zebrafish skeletal muscle does not show any transcription of ANO2 RNA.

The authors demonstrated a Cl current activated by SR Ca release that depends on external Cl. This was not present in the DHPR β 1-null zebrafish mutant relaxed and was restored by SR Ca depletion with caffeine. They also describe voltage and Ca dependence of presumed ANO1 current. While this looks and feels like ANO1, ultimately the authors should use CRISPR editing to knockout ANO1 a or b to directly demonstrate their roles.

We agree with the reviewer that while all the features “look and feel like ANO1” the final and direct proof of the functional expression of ANO1 channels in zebrafish skeletal muscle is the demonstration of abolishment (or significant reduction) of these specific Ca^{2+} -activated Cl^- currents by ANO1 knock out or knock down. We also agree with the reviewer that CRISPR/Cas editing would be one of the methods to reach this aim. However, on one hand to stay in time with the experimental requirements for the revision of our manuscript and on the other hand to be able to fluorescence-tag ANO1 knock down myotubes in order to identify them in patch-clamp experiments (discussed in our 1st answer to reviewer #1), we implemented a short hairpin RNA (shRNA) knock down strategy, in which along with short interfering RNAs (siRNAs), CFP is also expressed. As also lined out in our 1st answers to reviewers #1 and #3, we designed a series of shRNA constructs targeted against both ANO1 isoforms and used an approach shown to work well in zebrafish (*Dong et al., PLOS ONE, 2009; De Rienzo et al., Zebrafish, 2012*). From the 10 siRNAs that were expressed in normal zebrafish myotubes and tested in patch-clamp experiments for the knock-down of Ca^{2+} -activated Cl^- currents, 7 showed significant current reduction compared to control scrambled siRNA. These experiments finally link the presence of ANO1 to Ca^{2+} -activated Cl^- conductance in zebrafish skeletal muscle.

These siRNA knock down experiments are exemplified by recordings of Cl^- currents and Ca^{2+} release following the expression of the most active siRNA (with a homology of 91% between ANO1-a and ANO1-b) and are represented in Figure 4 and explicated in the Methods (lines 348 - 368) and Results section (lines 148 - 162) of the revised manuscript.

The muscle action potential was longer in relaxed mutants and with $\text{Cl}^- = 0$, suggesting a role for ANO1 in regulating spike width in zebrafish skeletal muscle, as it does in other cell types. But the currents in Fig 7b are brief – is this a different time scale than 7a?

The reviewer is right that the muscle action potential is longer in the mutant *relaxed* under physiological Cl^- concentration as well as in normal myotubes with $\text{Cl}^- = 0$. In both the cases (Fig 8a and Fig 8b) [new figure numbers in the revised manuscript!] the injection currents, indicated by small pedestals below the AP traces, are identical (3.8 nA for 1-ms). Moreover, the time scale for both recordings in Fig 8a and Fig 8b is alike.

The authors show build up of depolarization in relaxed mutants but not in normal muscle. Is this seen in normal muscle with $\text{Cl}^- = 0$? Of greater physiological relevance, does this occur upon motoneuron.

We performed these spike-train experiments in *relaxed* myotubes under normal external Cl^- conditions (142 mM) to avoid functional blockade of the canonical skeletal muscle Cl^- channel CIC-1 and all other putative Cl^- channels and pumps under $\text{Cl}^- = 0$ conditions, in addition to the ANO1 channel. In our opinion, these unspecific blocking effects would lead to an ambiguous

experimental outcome, compared to the clear-cut answer resulting from exclusive targeting ANO1 due to the lack of Ca²⁺ release-activation of ANO1 in *relaxed* myotubes.

Our answer to the second part of the reviewer's comment: The lack of the ANO1 current in *relaxed* myotubes does not affect motorneuron action, because structure and function of motoneurons were previously described to be fully intact in this paralytic mutant zebrafish *relaxed* (Ono et al., *J. Neurosci.*, 2001).

Finally, on a technical point, the mutant shows gradual depolarization and not a 'plateau' potential as the latter is a specific term eg for sustained depolarization in the absence of stimuli that is eliminated by strong, brief hyperpolarization, among other features.

The original term "plateau potential" was derived from the plateau phase of the action potential in the vertebrate heart (Weidmann, *J. Physiol. Lond.*, 1951). Later it was not only used for biophysical features as described by the reviewer (Fig. 1 in Masurkar & Chen, *Neurosci.*, 2011) but also for the rise of the baseline potential in a train of spikes with prolonged depolarizations (Fig. 4a in Gunaydin et al., *Nature Neurosci.*, 2010) like in our study. However, in agreement with the reviewer we exchanged the term "plateau potential" with the new term "depolarised plateau" which was earlier used to describe a train of APs that show a rise of the baseline potential due to the slowing down of individual APs by DAPs (Fig. 2 in Kim et al., *J. Neurosci.*, 2010), a feature quite comparable as seen in our recordings. The changes are indicated in the Results section (lines 293, 295 - 296) as well as in Fig. 9 and the corresponding Figure legend (lines 724, 725, 727, and 730) of our revised manuscript.

Our answers to the comments of **Reviewer #3**:

The manuscript by Dayal, Ng and Grabner employs a sophisticated combination of molecular, immunocytochemical and biophysical approaches to characterize calcium-activated chloride channel expression, subcellular localization and function in zebrafish skeletal muscle.

Using isoform-specific RT-PCR primers, the authors show that ANO1-a is predominantly expressed in red muscle, while ANO1-b is predominantly expressed in white muscle. Immunocytochemical studies indicate that ANO1 channels primarily localize to the surface membrane and not in the transverse tubule membrane. Whole cell patch clamp studies in myotubes from normal and relaxed zebrafish (which lack depolarization-induced elevations in intracellular calcium) provide convincing evidence for a robust chloride conductance consistent with ANO1 activity that requires an increase in cytosolic calcium. Specifically, the authors show the current is abolished by removing extracellular chloride and is absent in myotubes from relaxed zebrafish. Importantly, robust chloride currents could still be

elicited in relaxed myotubes during application of 8 mM caffeine to release calcium from intracellular stores (though calcium levels were not measured during these experiments). By dialyzing myotubes with known concentrations of free calcium, the authors show that the chloride current exhibits outward rectification at low calcium levels and linear voltage-dependence at cytosolic high calcium levels (13 μ M), consistent with prior measurements of ANO1 chloride currents. Current clamp studies in myotubes from normal and relaxed zebrafish revealed that the calcium-activated chloride current significantly reduces action potential duration, such that trains of action potentials with a stable negative plateau potential are possible up to frequencies as high as 80 Hz. Thus, while chloride channel function plays a similar role in maintaining normal muscle electrical excitability in both zebrafish and mammals, this is accomplished through two entirely different channels (ANO1 vs CLC1) with the relevant chloride conductances operating across vastly different voltage ranges (depolarized vs hyperpolarized potentials).

This study provides compelling evidence for a robust calcium-dependent chloride conductance in zebrafish skeletal muscle required to maintain electrical stability during high frequency tetanic contractions that is likely crucial for aquatic predator-prey swimming behaviors. In addition, the study provides convincing evidence for another intriguing evolutionary/physiological difference between euteleost and mammalian skeletal muscle. The experiments are carefully performed, appropriately analyzed and the manuscript is well-written.

I only have several relatively minor suggestions, which should help to further strengthen the conclusions and overall impact of this study.

1) Together, the ANO1 mRNA expression and biophysical properties of calcium-activated chloride channel function in zebrafish muscle are strongly suggestive that the currents measured are due to ANO1 channels. However, the results are correlative and definitive data linking the two are lacking. The final nail in the coffin would be to show that the chloride currents are abolished by morpholino-induced ANO1 knockdown. Morpholino ANO1 knockdown studies would also enable addressing another concern regarding the specificity of the antibody used in the immunocytochemical analysis of ANO1 subcellular localization. Fig. 4 presents a single representative image showing ANO1 localization to the surface membrane. Does this antibody only recognize ANO1 in zebrafish muscle? Does a western blot of zebrafish muscle using this antibody show only one band at the expected molecular weight for ANO1 or are multiple proteins recognized by this antibody? Is a similar pattern shown using multiple different ANO1 antibodies? All of these questions could be addressed by showing that the immunofluorescence signal in the surface membrane is abolished following morpholino-induced ANO1 knockdown.

The reviewer is correct that the final and direct proof of the functional expression of ANO1 channels in zebrafish skeletal muscle would be the demonstration that these specific Ca^{2+} -activated Cl^- currents can be abolished (or significantly reduced) by ANO1 knock-down. We also agree with the reviewer that morpholino-induced ANO1 knock down would be among one of the promising methods to reach this goal. However, with the aim to fluorescence-tag ANO1 knock-down myotubes to be able to identify them in patch-clamp experiments (see our 1st answer to reviewer #1), we implemented a short hairpin RNA (shRNA) knock-down strategy, where in parallel to the short interfering RNAs (siRNAs) also CFP is expressed. Hence, we designed a series of shRNA constructs targeted against both ANO1 isoforms and used a procedure shown to work well in zebrafish (*Dong et al., PLOS ONE, 2009; De Rienzo et al., Zebrafish, 2012*). Out of the 10 siRNAs that were expressed in normal zebrafish myotubes for the knock down of Ca^{2+} -activated Cl^- currents and tested in patch-clamp experiments, 7 showed significant CaCC current reduction compared to the control scrambled shRNA. These experiments finally link the presence of ANO1 to Ca^{2+} -activated Cl^- conductance in zebrafish skeletal muscle.

These siRNA knock down experiments are exemplified as recordings of CaCC currents and SR Ca^{2+} release following expression of the most active shRNA (with a homology of 91% between ANO1-a and ANO1-b) and are represented in Figure 4 and explicated in the Methods (lines 348 - 368) as well as in the Results (lines 148 - 162) sections of the revised manuscript. In full agreement with the reviewer's assessment that ANO1 knock down studies would also address concerns regarding the specificity of the antibody used for analysing the subcellular localization of ANO1 (Figure 5), we performed immunofluorescence assays on siRNA expressing myotubes. And indeed we saw a significant reduction of the immunofluorescence signal in the surface membrane of myotubes following siRNA-induced ANO1 knockdown (demonstrated in Supplementary Figure 3 and discussed in the Results section (lines 182 - 189) of the revised manuscript). Consequently, we can conclude that the antibody used is specific for ANO1 and as nicely exemplified by the reviewer answer in advance all these kind of theoretical questions.

2) From the results in Fig. 6 (particularly those from relaxed myotubes), the authors conclude that a low level of CLC1 current is present in zebrafish muscle. However, this current exhibits a linear current-voltage relationship rather than the prominent inward rectification that is emblematic of CLC1 currents. In order to support the authors' conclusion that this current is indeed due to CLC1 channels, the authors should demonstrate that the current is blocked by a CLC1 channel inhibitor (e.g. 9-anthracene carboxylic acid) and/or morpholino-induced knockdown of CLC1.

As suggested by the reviewer, we tried to block the "low level" Cl^- current with the CLC-1 channel inhibitor 9-anthracene carboxylic acid (9AC) to strengthen our conclusion that it is

indeed Cl⁻ flux through the skeletal muscle CIC channel. The resulting significant reduction of the Cl⁻ current in the presence of 9AC clearly and directly confirms that the Cl⁻ current in *relaxed* myotubes is actually due to CIC conductance. Results are depicted as Supplementary Figure 5 and elucidated in the Methods (lines 406 - 408) and Results sections (lines 238 - 240) of the revised manuscript: “....we observed a dramatic reduction in peak CIC current density (77% at -140 mV and 66% at +60 mV) in the presence of 1 mM 9AC, a blocker of CIC-1 channels⁵⁷ (Supplementary Fig. 5).”

The reviewer stated that the zebrafish CIC current exhibits a linear current-voltage relationship rather than the prominent inward rectification that is emblematic of CIC-1 currents. In this respect we would like to mention that the observed inward rectification of CIC currents is primarily due to the non-physiologically high Cl⁻ concentrations of internal solutions used in most electrophysiological studies on CIC. Even though, intracellular Cl⁻ concentrations in skeletal muscle were reportedly between ~5 mM in frog (Hironaka & Morimoto, *Jpn. J. Physiol.*, 1980) and ~9 mM in human (Kowalchuk *et al.*, *J. Appl. Physiol.*, 1988), internal solutions used for patch clamp recordings of CIC commonly contained Cl⁻ concentrations between 40 mM (Lueck *et al.*, *J. Gen. Phys.*, 2010) to more than 100 mM (Falke *et al.*, *Neuron*, 1995). With this in mind, we used in our present study the more physiological Cl⁻ concentration of 4 mM for the internal recording solution. Moreover, previous patch clamp recordings on mouse skeletal muscle revealed that CIC current kinetics is highly dependent on the Cl⁻ concentration of the internal solution ([Cl⁻]_i) (DiFranco *et al.*, *J. Gen. Phys.*, 2011). Inward rectification is pronounced at a [Cl⁻]_i of 70 mM but is considerably weaker at 40 mM. Interestingly, at a [Cl⁻]_i of 10 mM CIC currents lose their inward rectification and exhibit a linear current-voltage relationship, comparable to what we found in zebrafish

3) The recordings of action potential trains in normal and relaxed myotubes in Fig. 8 are elegant. The authors' conclusion that the differences observed in action potential plateau levels are vital for maintaining high speed contractile behavior would be strengthened by showing enhanced or more sustained calcium transient amplitudes during repetitive 80 Hz stimulation trains in myotubes from normal fish compared to that obtained from relaxed fish.

We fully agree with the reviewer that whenever a conclusion or statement can be strengthened by additional experiments, efforts should be made accordingly. Unfortunately, in this special case it is not possible to test normal myotubes in comparison to *relaxed* myotubes for enhanced or more sustained Ca²⁺ transients at 80 Hz stimulations because, as demonstrated in Fig. 1c, d, *relaxed* myotubes not only lack the ANO1 Cl⁻ current (left panels) but also fully lack Ca²⁺ release transients (right panels).

4) Page 3, line 57. In smooth muscle cells, the intracellular chloride concentration is not higher than that of the extracellular space (e.g. see Table 1 in review of Chipperfield and Harper, 2000).

As pointed out by the reviewer, we corrected the statement as follows: (lines 57 - 58 in the revised manuscript): “.....the intracellular Cl^- concentration in smooth muscle cells is high due to active accumulation by Cl^-/HCO_3^- exchange and Na^+ , K^+ , Cl^- co-transportation¹¹”. In addition, we replaced citation #11 with the more accurate paper of *Bulley & Jaggard, Pflügers Arch., (2014)*.

5) Given the results in Fig. 3f (DHPR data shows 24% contamination of red muscle while ANO1-b is 45% in red muscle), it is too strong to state in the results section that ANO1-a is “exclusively” expressed in red muscle (and ANO1-b in white muscle). In the absence of more definitive data, it would be better to stick with the terminology used in the legend to Figure 3 (“predominantly” or “mainly”).

As suggested by the reviewer we exchanged the term “*exclusively*” to “*predominantly*” (line 147 in the revised manuscript).

6) Page 12, line 261. While 50 Hz is nearly sufficient for tetanus in soleus, a much higher stimulus frequency (125-150 Hz) is actually required to achieve tetanus in fast twitch skeletal muscle.

There is a big discrepancy in different reports regarding the tetanic fusion frequency. Especially in the older literature, due to technical limitations and a theoretical constrain, the values seem to be quite overestimated. The fact is, that it is problematic or even theoretically impossible to accurately pinpoint the exact stimulation frequency at the onset of a complete tetanus i.e., when the fusion index [FI] reaches 100%, because FI-frequency curves are S-shaped (asymptotic). This theoretical constrain was mathematically assessed in the interesting paper of *Watanabe et al., J. Electromyogr. Kinesiol., (2010)*. In accordance with this study we avoided expressing the asymptotic increasing frequency values as 100% for the complete tetanus formation, as well as, for maximum force generation, but used scientifically more accurate 90% values.

Accordingly, in the revised manuscript we modified the text (lines 281 - 286) as follows: “.....”
Muscle contractions start to sum up beyond stimulation frequencies of 5 - 20 Hz until the response forms a smooth ramped increase of tetanic contraction, which - depending on species and muscle type - reaches 90% of its maximum fusion rate (complete tetanus) at 20 - 60 Hz. Increased stimulation frequency leads to increased force production and a maximum force of 90% is generated around 60 - 80 Hz²⁸⁻³¹. The usual firing rate of vertebrate motor neurons during voluntary muscle contraction is within this tetanic range³².”

In addition, we replaced the former references #27, #28 and #30 with the more accurate papers of *Watanabe et al. (2010)*, *Mròwczynski et al. (2006)*, *Monster & Chan (1977)*, and

Altringham & Johnston (1988), which more clearly support our quoted stimulation frequency values and corresponding statements in the revised paragraph.

7) Reference 34. The last name of the first author of the intended reference is Davis, not Alison.

As pointed out by the reviewer, the name of the first author has been corrected to *Davis, A. J. et al.* (reference #35 in the revised manuscript).

8) The authors go overboard touting the purported phylogenic superiority of euteleost fish. There are at least 7 or 8 statements similar to “euteleost fishes is evolutionary highly advanced compared to mammals.” The statements are made so often that it becomes “preachy.” It should be sufficient to make this assertion once or twice at key places in the manuscript.

Incited by frequent questions and comments of academic colleagues, who wondered why several features (e.g., non-Ca²⁺ conductance of the DHPR or expression of ANO1 in skeletal muscle) were already developed in the “most primitive” vertebrate species “fish” but later got lost on the phylogenetic path to the “highest evolved” mammals, we might have overdone with the repetition of this evolutionary fact in our manuscript. We thank the reviewer for pointing this out and as suggested, we either removed these statements or modified them. Hence, in the revised manuscript this assertion is made only in the Abstract (line 9), the Introduction (lines 37 - 38), and the concluding paragraph (lines 318 - 319).

Reviewers' Comments:

Reviewer #1:

Remarks to the Author:

A really, really elegant paper--well done!

Reviewer #2:

Remarks to the Author:

The authors have satisfactorily addressed my questions and performed important new experiments, such as knockdown of ANO1.

Reviewer #3:

Remarks to the Author:

The revised manuscript and Author Response Letter fully address all concerns raised in my review. I have no further concerns.

Response to the Referees:

REVIEWERS' COMMENTS:

Reviewer #1 (Remarks to the Author):

A really, really elegant paper--well done!

Reviewer #2 (Remarks to the Author):

The authors have satisfactorily addressed my questions and performed important new experiments, such as knockdown of ANO1.

Reviewer #3 (Remarks to the Author):

The revised manuscript and Author Response Letter fully address all concerns raised in my review. I have no further concerns.

We sincerely thank our reviewers for the overall constructive and valuable input.